# EMT, Stemness, and Drug Resistance in Biological Context: A 3D Tumor Tissue/In Silico Platform for Analysis of Combinatorial Treatment in NSCLC with Aggressive KRAS-Biomarker Signatures

**DOI:** 10.3390/cancers14092176

**Published:** 2022-04-27

**Authors:** Matthias Peindl, Claudia Göttlich, Samantha Crouch, Niklas Hoff, Tamara Lüttgens, Franziska Schmitt, Jesús Guillermo Nieves Pereira, Celina May, Anna Schliermann, Corinna Kronenthaler, Danjouma Cheufou, Simone Reu-Hofer, Andreas Rosenwald, Elena Weigl, Thorsten Walles, Julia Schüler, Thomas Dandekar, Sarah Nietzer, Gudrun Dandekar

**Affiliations:** 1Chair of Tissue Engineering and Regenerative Medicine, University Hospital Würzburg, Röntgenring 11, 97070 Würzburg, Germany; matthias.peindl@uni-wuerzburg.de (M.P.); claudia.goettlich@crl.com (C.G.); nto.hoff@gmail.com (N.H.); tamara.luettgens@stud-mail.uni-wuerzburg.de (T.L.); schmitt_franziska@gmx.de (F.S.); jesus.nieves@uni-wuerzburg.de (J.G.N.P.); celina.may@stud-mail.uni-wuerzburg.de (C.M.); anna.schliermann@gmail.com (A.S.); ckronenthaler@aol.com (C.K.); elena.weigl@med.uni-muenchen.de (E.W.); sarah.nietzer@uni-wuerzburg.de (S.N.); 2Department of Bioinformatics, Biocenter, University of Würzburg, Am Hubland, 97074 Würzburg, Germany; samantha.crouch@uni-wuerzburg.de; 3Department of Thoracic Surgery, Klinikum Würzburg Mitte gGmbH, Salvatorstr. 7, 97074 Würzburg, Germany; danjouma.cheufou@kwm-klinikum.de; 4Department of Pathology, University of Würzburg, Josef-Schneider-Str. 2, 97080 Würzburg, Germany; simone.reu@uni-wuerzburg.de (S.R.-H.); rosenwald@uni-wuerzburg.de (A.R.); 5Comprehensive Cancer Center Mainfranken, Josef-Schneider-Straße 6, Building C16, 97080 Würzburg, Germany; 6Department of Thoracic Surgery, University Medicine Magdeburg, Leipziger Straße 44, 39120 Magdeburg, Germany; thorsten.walles@med.ovgu.de; 7Charles River Discovery Research Services Germany GmbH, Am Flughafen, 14, 79108 Freiburg, Germany; julia.schueler@crl.com; 8European Molecular Biology Laboratory (EMBL) Heidelberg, Structural and Computational Biology, Meyerhofstraße 1, 69117 Heidelberg, Germany; 9Fraunhofer Institute for Silicate Research (ISC), Translational Center Regenerative Therapies, Röntgenring 11, 97070 Würzburg, Germany

**Keywords:** EMT, drug resistance, invasion, stemness, 3D lung tumor tissue models, KRAS biomarker signatures, boolean in silico models, targeted combination therapy

## Abstract

**Simple Summary:**

The phenotypic transition of tumor cells from epithelial to mesenchymal characteristics is called EMT and is widely discussed in the scientific community as a game changer in drug resistance and metastasis formation. However, clinical studies could not prove the efficacy of EMT-interfering treatments, and in clinical routine, EMT is not investigated to assess invasion. To fill this gap between bench and bedside, we use in this study a lung tumor tissue model with a preserved basement membrane for investigation of EMT functions with respect to invasion across this membrane and drug resistance. Our results suggest EMT is more a marker of drug resistance than a maker. Invasion is enhanced by EMT but more dependent on intrinsic factors, and EMT is not detected in the center of invasive tumor nodules. An in silico signaling network model is used to integrate these in vitro results and to reveal determinants for drug response.

**Abstract:**

Epithelial-to-mesenchymal transition (EMT) is discussed to be centrally involved in invasion, stemness, and drug resistance. Experimental models to evaluate this process in its biological complexity are limited. To shed light on EMT impact and test drug response more reliably, we use a lung tumor test system based on a decellularized intestinal matrix showing more in vivo-like proliferation levels and enhanced expression of clinical markers and carcinogenesis-related genes. In our models, we found evidence for a correlation of EMT with drug resistance in primary and secondary resistant cells harboring *KRAS^G12C^* or *EGFR* mutations, which was simulated in silico based on an optimized signaling network topology. Notably, drug resistance did not correlate with EMT status in *KRAS*-mutated patient-derived xenograft (PDX) cell lines, and drug efficacy was not affected by EMT induction via TGF-β. To investigate further determinants of drug response, we tested several drugs in combination with a KRAS^G12C^ inhibitor in *KRAS^G12C^* mutant HCC44 models, which, besides EMT, display mutations in *P53*, *LKB1*, *KEAP1,* and high c-MYC expression. We identified an aurora-kinase A (AURKA) inhibitor as the most promising candidate. In our network, AURKA is a centrally linked hub to EMT, proliferation, apoptosis, LKB1, and c-MYC. This exemplifies our systemic analysis approach for clinical translation of biomarker signatures.

## 1. Introduction

The cellular program of epithelial-to-mesenchymal transition (EMT) and its dynamic nature is believed in the tumor biology community to be of great importance for the development of strategies to fight advanced and therapy-resistant cancer [1]. Entering the keywords “epithelial mesenchymal transition cancer” in Pubmed (https://pubmed.ncbi.nlm.nih.gov, accessed on 1 December 2021) reveals nearly 30,000 publications starting in 1975 and gaining exponential growth about 10 years ago. Adding the term “invasion” results in over 15,000 publications or “resistance” in over 5000. As a major player, TGF-β is suggested to induce not only EMT but also drug resistance [2,3], stressed by over 400 publications starting from 2002 when adding “TGF” to “resistance”. As an approach going beyond EMT, the shut-down of cellular plasticity was suggested, which is illustrated by a search result of about 1500 publications since 2009 when adding the search term “stemness” to “EMT” and “cancer” in Pubmed. Regarding lung cancer, there has been a rising number of publications for about 10 years on the topic of EMT (in total, about 3700). A vast amount of work has been performed to translate EMT-related scientific results into the clinic with limited success. Several approaches to revert EMT to mesenchymal-to-epithelial transition (MET) ended up disappointingly in clinical trials (overview: the work of [4]). One conclusion was that EMT is a metastable stage, and therapeutic intervention strategies will lead to the activation of compensatory pathways to regain homeostasis and resistance. From the clinical perspective, EMT markers are not investigated except for colorectal cancer (CRC), where the translocation of β-catenin toward the nucleus is one hallmark of cancer progression. In the nucleus, β-catenin induces the expression of genes responsible for EMT induction and stemness [5]. However, a deep understanding of EMT and its correlation with invasion, stemness, and drug response and interdependent signaling pathways related to biomarkers is still needed for an effective translation into the clinic.

For non-small cell lung carcinomas (NSCLC), especially adenocarcinomas, biomarker-guided therapies were successfully introduced into patient care about 17 years ago, mostly targeting activating *EGFR* mutations [6]. However, secondary resistance often arises after a few months of treatment, and the largest group of patients suffering from tumors carrying a *KRAS* mutation even shows high primary resistance to targeted treatment. For NSCLCs harboring *KRAS^G12C^* mutations (representing more than 40% of the most frequent *KRAS* mutation in lung cancer), sotorasib, an allele-specific covalent inhibitor, received accelerated approval by the FDA in May 2021 and showed efficacy in certain subgroups during clinical trials [7,8]. Combinatorial treatments are under investigation [9]. The identification of frequent co-mutations paves the way for understanding mechanistically how to improve tailored therapies also in resistant groups [10]. A major restriction for effective translation is the availability of models including in vivo-like biological features capable of reliably evaluating drug efficacy. This is illustrated by the high attrition rates of existing preclinical models [11,12].

As a solid basis for the simulation of tumor biological mechanisms in vitro, we use a tissue-engineered tumor model based on an intestinal decellularized tissue matrix termed SISmuc (small intestinal submucosa with preserved mucosa). It allows tumor cell growth to reach homeostasis with changed expression patterns compared to 2D cultures, especially regarding markers of proliferation and apoptosis [13], but also (as demonstrated here) regarding carcinogenesis-related and clinical markers. Notably, our model shows a higher chemoresistance and a better predictivity compared to 2D and animal experiments in several biomarker-guided targeted test set-ups [14,15]. Furthermore, it allows discrimination between functional non-invasive and invasive cells due to its preserved basement membrane, which has been shown to be crossed by cells after EMT induction via TGF-β [16,17].

In this study, to gain insight into dependencies regarding EMT, invasion, stemness, and drug resistance, we induced EMT by TGF-β1 in different lung cancer cell lines grown in 3D tissue models and then treated those models with biomarker-corresponding targeted therapies. We analyzed changes in tissue morphology (cell invasion), differentiation (E-cadherin, muc-1), EMT (cytokeratins, vimentin), and stemness (CD44). While we could indeed demonstrate a higher EMT status in secondary resistant *EGFR*-mutated cells, which was also true for primary more resistant *KRAS*-mutated cells, no increase in resistance after induction of EMT via TGF-β1 was observed. Moreover, in four *KRAS^G12C^*-mutated patient-derived xenograft (PDX)-derived cell lines, neither EMT status nor stemness consistently correlated with resistance in 2D and 3D cell culture. The environment’s intrinsic invasion properties surpass TGF-β1 induced, and thereby EMT-related, invasion. Evaluation of clinical lung adenocarcinoma specimens further underlines an EMT-independent invasion mechanism at least in the center of invasive tumor nodules. Changes in drug response under 3D culture conditions stress the high impact of the tissue environment on preclinical testing results.

Our central hypothesis in this paper is that EMT is not a consistent predictor of drug response. Instead, we propose that it is the individual biomarker signature of a cell line that allows for predicting the effectiveness of individualized therapies targeting tumor-specific cascades in a complex network. A combined in vitro/in silico modeling approach was hence pursued to understand the complex pathway interdependencies in the cellular network.

To improve the understanding of resistance mechanisms, we in silico simulated the sensitivity of the more epithelial *KRAS^G12C^*-mutated cell line H358 and the higher resistance of the more mesenchymal *KRAS^G12C^*-mutated cell line HCC44 toward a KRAS inhibitor. Before the simulation, we updated our previously established in silico topology of a cancer signaling network [14,15,16,17] by integrating frequent co-mutations from patients with *KRAS*-mutated lung cancer [10] and modified it in a cancer cell line-specific way. After incorporating experimental data, we had a cell type-specific model. We tested the KRAS inhibitor in several combination therapies in vitro and in silico as suggested by large screening experiments employing the more resistant HCC44 cell line exhibiting high EMT status, CD44 and c-MYC expression, and other common *KRAS* co-mutations. After adjusting the in silico model to three different in vitro combination treatments, we could, as proof of concept, reveal the combination of an aurora-kinase A (AURKA) inhibitor together with a KRAS^G12C^ inhibitor to be effective, predicted simultaneously by both in silico and in vitro experiments.

The present tissue-based model combined with in silico analysis should pave the way for a better tumor biological understanding of co-mutations, EMT, stemness, and drug resistance and reveal biomarker signatures for patient stratification in clinical studies.

## 2. Materials and Methods

### 2.1. Matrix Preparation

Chemical decellularization of jejunal segments was performed as previously published [18,19,20]: the intestinal segments were explanted, rinsed, and chemically decellularized with a sodium deoxycholate monohydrate solution; the vascular tree was manually removed; and the whole product underwent gamma sterilization. Tumor models were prepared as published before [13].

### 2.2. Cells

All cells were cultured under standard culture conditions (37 °C, 5% CO_2_). PDX-derived cell lines LXFA 983, LXFL 1072, LXFL 1674, and LXFA 2184 were kindly provided by Dr. Julia Schüler (Charles River, Freiburg, Germany). PDX-derived cell lines, A549 (purchased from DSMZ), H358 (purchased from ATCC), and HCC44 cells (purchased from DSMZ) were cultured in RPMI 1640 medium with GlutaMAX™ (Gibco/life technologies/ThermoFisher Scientific, Waltham, MA, USA) supplemented with 10% FCS (PAN-Biotech, Aidenbach, Germany). HCC827 cells (purchased from DSMZ) were cultured in RPMI 1640 medium with GlutaMAX™ (Gibco/Life Technologies/ThermoFisher Scientific, Waltham, MA, USA) supplemented with 20% FCS (PAN-Biotech, Aidenbach, Germany). HCCresA1, HCCresA2, and HCCresA3 were generated from the cell line HCC827 via the constant addition of 1 µM gefitinib (Selleckchem, Houston, TX, USA) to the cell culture medium. While HCCresA2 and HCCresA3 cells displayed resistance toward the treatment with gefitinib in concentrations up to 10 µM, HCCresA1 cells still exhibited an intermediate sensitivity in comparison (Appendix A). HCCresA2 and HCCresA3 cells displayed reduced sensitivity toward EGFR inhibition after about 4 months of permanent treatment with 1 µM gefitinib, whereas HCCresA1 still showed a partial sensitivity in this time period. Regarding gefitinib sensitivity, we are referring to Noro et al. cells with an IC_50_ lower than 1 µM as highly sensitive, cells with an IC_50_ between 1 and 10 µM as intermediately sensitive and cells with an IC_50_ greater than or equal to 10 µM as resistant, respectively [21]. Primary human lung fibroblasts were isolated from biopsies of healthy lung tissue with informed consent according to ethical approval granted by the Institutional Ethics Committee of the University Hospital Würzburg (protocol code 99/20-am) and cultured in DMEM with GlutaMAX^TM^ (Gibco/life technologies/ThermoFisher Scientific, Waltham, MA, USA) supplemented with 100 mM sodium pyruvate and 10% FCS (PAN-Biotech, Aidenbach, Germany).

### 2.3. Preparation of Tumor Models

Single pieces of the SISmuc matrix were fixed between two metal rings (cell crowns) and seeded with 100,000 tumor cells on the mucosal side (transwell in Figure 1A). Tumor models were placed in 12-well plates with 1 mL media in the inner compartment and 1.5 mL in the outer compartment of the cell crown. For static cell culture, the models were cultured for 11 days and afterward treated with the test substance for 3 days. During static and semi-dynamic cell culture, medium was changed every 2 to 3 days. All test substances were administered via the cell culture medium. Here, A549, H358, HCC44, and HCC827 tumor models were cultured statically, while PDX-derived cell lines tumor models were grown under semi-dynamic conditions on an orbital shaker at 100 rpm until day 10 of culture before the inhibitor treatment started.

### 2.4. Patient Tumor Samples

Human tracheo-bronchial tissue samples for staining were obtained from adult patients undergoing elective pulmonary resection at the University Hospital Würzburg, the Hospital Würzburg Mitte gGmbH, and the University Hospital Magdeburg. Written informed consent was obtained beforehand, and the studies were approved by the institutional ethics committees on human research of the Julius-Maximilians-University Würzburg (protocol code 215/12 and 99/20-am) and Otto-von-Guericke University Magdeburg (protocol code 163/17), respectively. Patient material was derived from 10 NSCLC adenocarcinoma patients older than 18 years from both sexes. Ethnicity, previous treatments, and other medical disorders were neglected.

### 2.5. Treatment of Cells in 2D and 3D

For 2D cell culture, cells were cultured in 12-, 24-, or 96-well plates for subsequent M30 Cyto Death™ ELISAs (PEVIVA^®^, TECOmedical, Sissach, CH), immunofluorescence stainings, or CellTiter-Glo^®^ viability assays (Promega, Madison, WI, USA), respectively. Inhibitors dissolved in DMSO (Sigma-Aldrich, München, Germany) or water were applied in different concentrations to the cell culture medium for 3 days, 24 h after seeding the cells, with a medium change on the 2nd day of treatment. For 3D cell culture, cell lines were seeded on the matrix SISmuc and cultured for 11 days prior to adding medium containing the corresponding inhibitors. Medium containing the inhibitors was renewed after 48 h of treatment. The following inhibitors were used to treat the cells in 2D and 3D cell culture: alisertib (Selleckchem, Houston, TX, USA), ARS-1620 (Hycultec, Beutelsbach, Germany), crizotinib (Selleckchem, Houston, TX, USA), erdafitinib (Selleckchem, Houston, TX, USA), gefitinib (Selleckchem, Houston, TX, USA), metformin HCl (Selleckchem, Houston, TX, USA), and SHP099 HCl (Selleckchem, Houston, TX, USA).

### 2.6. Stimulation of Cells with hTGF-β1

Cells were seeded on the SISmuc as described above. After 3 days in culture, medium containing 2 ng/mL hTGF-β1 with a carrier (Cell Signaling, Danvers, MA, USA) was added to the models and renewed every 2 to 3 days for the remaining 11 days in culture.

### 2.7. Immuno/Histochemical Stainings

Tumor models were fixed in a 4% PFA solution (Carl Roth GmbH, Karlsruhe, Germany) for 2 h, embedded in paraffin, and cut in a microtome at 3–5 µm. Hematoxylin and Eosin (H&E) staining (Morphisto, Offenbach am Main, Germany) was performed according to the manufacturer’s protocol. Non-immunofluorescent immunohistochemical stainings were performed using the 3,3′-Diaminobenzidine (DAB) system (DCS Innovative Diagnostik-Systeme, Hamburg, Germany) according to the manufacturer’s protocol resulting in the formation of a brown dye at areas of antibody binding. The following primary antibodies were used: rabbit anti-TTF1 (Abcam Cat# ab76013, RRID: AB_1310784, Cambridge, UK), rabbit anti-SPP1 (Abcam Cat# ab8448, RRID: AB_306566), mouse anti-Cytokeratin 7 (Abcam Cat# ab9021, RRID: AB_306947), rabbit anti-P63 (Abcam Cat# ab53039, RRID: AB_881860), rabbit anti-Collagen IV (Abcam Cat# ab6586, RRID: AB_305584), rabbit anti-vimentin (Abcam Cat# ab92547, RRID: AB_10562134), rabbit anti-Ki-67 (Abcam Cat# ab16667, RRID: AB_302459), rabbit anti-Mucin-1 (Abcam, Cat #ab84597, RRID_ AB_10672326), mouse anti-E-cadherin (BD-Biosciences, Cat#61081, RRID:AB_397581), rabbit anti-CD44 (Abcam, Cat#ab51037, RRID: AB_868936), rabbit anti-β catenin (Abcam, Cat#ab32572, RRID: AB_725966), and mouse anti-Cytokeratin, pan (Sigma-Aldrich Cat# C2562, RRID: AB_476839). Primary antibodies were diluted 1:100 in antibody diluent (DCS Innovative Diagnostik-. Systeme, Hamburg, Germany) and incubated overnight at 4 °C. Secondary antibodies donkey anti-mouse IgG (Thermo Fisher Scientific Cat# A-31571, RRID: AB_162542) conjugated to Alexa-647 or donkey anti-rabbit IgG (ThermoFisher Scientific, Waltham, MA, USA Cat# A-31572, RRID: AB_162543) conjugated to Alexa-555 (ThermoFisher Scientific, Waltham, MA, USA) were diluted 1:400 in antibody diluent and incubated for 1 h at room temperature (RT). Cell nuclei were counterstained using 4′,6-diamidino-2-phenylindole (DAPI), which was diluted in the embedding medium Fluoromount-G (ThermoFisher Scientific, Waltham, MA, USA). Images were taken using a digital microscope (BZ-9000, Keyence, Osaka, Japan).

### 2.8. Quantification of Proliferation and Cell Invasion

For the determination of proliferation indices, Ki-67 stainings were quantified: at least 5 images of non-overlapping regions of each sample were taken with a fluorescence microscope (BZ-9000, Keyence, Osaka, Japan). DAPI and Ki-67-positive cells were counted manually using Image J (v1.53a, NIH, Bethesda, MD, USA). For the quantification of cell invasion, collagen IV immunofluorescence stainings were performed. Subsequently, DAPI-positive cells were manually counted on top and in the former crypts versus inside the biological matrix in at least 5 images of non-overlapping regions of each sample. For each image, the number of Ki-67-positive cells or cells inside the biological matrix was calculated in percent of total cell number, and the mean of each sample was used for subsequent statistical analysis.

### 2.9. M30 ELISA

The M30 CytoDeath™ (PEVIVA^®^, TECOmedical, Sissach, CH) or M30 ApoptoSense™ (PEVIVA^®^, TECOmedical, Sissach, CH) assay was used for the quantification of epithelial apoptosis by measuring the caspase-cleaved product of cytokeratin 18 in the cell culture supernatant. Both ELISAs were performed according to the manufacturer’s protocol. In short, supernatants of cells cultured in 12-well plates or the 3D tumor models were collected at 4 different time points: directly before and 24, 48, and 72 h after the inhibitor treatment. Samples in duplicates were diluted in the corresponding cell culture medium to fit the range of the standard curve, and the M30 conjugate was added to the samples in each well. After incubation on an orbital shaker for 3 h, wells were washed 5 times prior to the addition of the substrate. The reaction was stopped after 20 min. Absorbance was measured after shaking for 10 s with a microplate reader (TECAN, Männedorf, Switzerland). M30 quantifications were analyzed with Origin (OriginLab, Northampton, MA, USA). To calculate the fold increase in apoptosis after treatment, each sample was firstly normalized to its baseline increase in apoptosis in the last 24 h before the initial treatment. Subsequently, these normalized values of the respective timepoints (24, 48, or 72 h) after treatment were divided by the values of the untreated controls at the same time points.

### 2.10. Viability Assays (MTT-Test and CellTiter-Glo^®^)

To determine the viability of cells after inhibitor treatment in 2D cell culture, CellTiter-Glo^®^ Luminescent Viability assays (Promega, USA) were used according to the manufacturer’s protocol. In brief, cells were seeded in 96-well plates and were allowed to attach for 24 h. Subsequently, cells were treated for 72 h with the indicated inhibitors and washed once with PBS before the addition of CellTiter-Glo^®^ reagent diluted 1:2 in cell culture medium. Plates were mixed for 2 min and incubated at RT for 10 min before recording the luminescence with an integration time of 1000 milliseconds per well in a microplate reader (TECAN, Männedorf, Switzerland). IC_50_ values were calculated using Prism 8 (Graphpad Software, Inc., San Diego, CA, USA) by plotting the logarithmic concentrations versus the response with a variable slope. For the evaluation of viabilities after treatment in 3D tumor models, MTT assays were performed. Therefore, 3 mg/mL MTT (SERVA, Heidelberg, Germany) were diluted 1:3 in the corresponding cell culture medium and added to the models for 3 h at standard conditions. The biological matrix was removed from the cell crowns, and formazan was washed out of the SISmuc with 0.04 N HCl in isopropanol in 3 steps before measuring the absorbance at 570 nm with a microplate reader (Tecan, Männedorf, Switzerland).

### 2.11. Western Blotting

Tumor cells cultured on the SISmuc were lysed in lysis buffer (137 mM NaCl (Carl Roth GmbH, Karlsruhe, Germany), 20 mM Tris-HCl (Sigma Aldrich, München, Germany) pH 8.0, 2 mM EDTA (ThermoFisher Scientific, Waltham, MA, USA), 50 mM NaF (Sigma Aldrich, München, Germany), 1 mM NaVO_3_ (Sigma Aldrich, München, Germany), 10% glycerol (Carl Roth GmbH, Karlsruhe, Germany), 1% NP-40 (AppliChem GmbH, Darmstadt, Germany), 0.5% DCA (Carl Roth GmbH, Karlsruhe, Germany), 0.1% SDS (Carl Roth GmbH, Karlsruhe, Germany), 1× Protease Inhibitor (Roche, Munich, Germany)) for 1 h at 4 °C on a rocking shaker. For each sample, 80 µg protein was loaded per lane on a 10% SDS gel and subsequently blotted on a 0.2 µm nitrocellulose membrane. Blots were blocked for 1 h at RT in 5% milk in TBS-T (Sigma Aldrich, München, Germany). Primary antibodies against c-MYC or alpha-tubulin were incubated overnight at 4 °C in 5% BSA or 5% milk in TBS-T, respectively. Secondary antibodies were incubated in 5% milk in TBS-T at RT for 1 h. For the development of the blots, the WesternBright Chemilumineszenz Substrat Quantum kit (Biozym, Hessisch Oldendorf, Germany) was used and visualized on the Imaging Station FluorChemQ (Biozym, Hessisch Oldendorf, Germany). The following antibodies were used: rabbit anti-c-Myc (Y69) (Abcam Cat# ab32072, RRID: AB_731658), mouse anti-alpha-tubulin (DM1A) (Cell Signaling, #3873:, RRID: AB_1904178), goat anti-mouse IgG (H + L)-HRPO (Jackson Immuno Research, West Grove, PA, USA, #115-035-146, RRID: AB2307392) and goat anti-rabbit IgG (H + L)-HRPO ((Jackson Immuno Research, #111-035-144, RRID: AB_2307391). Quantitative evaluation was performed using ImageJ, and each sample was normalized to the loading control before comparison.

### 2.12. Statistics

Statistical significance was determined with Prism 8 (GraphPad, USA) using unpaired *t*-tests, assuming that data were distributed normally for 3 ≤ *n* ≤ 13. *p*-values ≤ 0.05 were considered significant; *: *p* ≤ 0.05, **: *p* ≤ 0.01, ***: *p* ≤ 0.001.

### 2.13. Lung Cancer PCR Array

For quantitative real-time PCR (qPCR), Human Lung Cancer PCR Array plates (PAHS-134D-12, Qiagen, Hilden, Germany) were used with the recommended RT2 SYBR Green Master Mix, according to the array’s protocol. cDNA was prepared using the iScript cDNA Synthesis Kit (Biorad, Hercules, CA, USA), and 20 μL of this cDNA sample was diluted 1:5 in H2O, subsequently added to 1350 μL of master mix and 1250 μL of water. This equals samples for 100 wells with cDNA from 10 ng of RNA per well. Of this master mix, 25 µL were pipetted into each well of a 96-well lung cancer PCR array plate (separate Appendix A). The qPCR was run as follows: 10 min at 95 °C, 15 s at 95 °C, 30 s at 55 °C repeated 40 times, 30 s at 72 °C, indefinitely at 4 °C. For analysis, the ΔΔCt value was determined, and the fold change 2^-ΔΔCt^ was calculated by dividing the normalized gene expression (2^-ΔCt^) of a test sample by the normalized gene expression of a control sample. Fold change was then transformed into fold regulation; fold changes greater than one were equal to the fold regulation, while fold changes smaller than one were inverted and presented as negative fold regulation values (e.g., fold change 0.5 equals a –2-fold regulation). All genes with a fold regulation greater than 3 or smaller than −3 and with at least one of the two compared Ct values < 30 were considered considerable transcription differences.

### 2.14. Ultrastructural Analysis

Samples were washed with pre-warmed PBS + calcium and magnesium (Sigma Aldrich, München, Germany). Cell-free edges were removed with a scalpel prior to fixation in a 6.25% or 2.5% solution of glutaraldehyde (Sigma Aldrich, München, Germany) overnight at 4 °C. Further sample preparation for raster electron microscopy (REM) as well as imaging was performed at the Imaging Core Facility, Biocenter, University of Würzburg.

### 2.15. In Silico 3D Tissue Simulations/Bioinformatics

To model individual drug actions as well as effects of drug combinations, dynamic simulations of cellular pathways also considered known impacts of different drugs. Signaling network reconstruction was based on available literature and on biochemical and human interactome database sources such as KEGG. For data-driven modeling, signaling network reconstruction was combined with dynamic simulations of cellular pathways: first, the network topology was created and edited in CellDesigner (version 3.5.1, The Systems Biology, Tokyo, Japan, accessed on: 1 February 2021; https://celldesigner.org/download351.html) [22] and exported as an xml file (separate Appendix A). Importantly, the Boolean logic of the network was considered, i.e., activating and inhibitory interactions between receptors, proteins, and protein cascades. Modifying crosstalk was also implemented. Next, dynamic simulations followed using the software SQUAD and using the network calculated to predict the activation or inhibition for every protein in the whole network as well as the resulting outcome parameters [23]. As kinetic information for individual protein nodes in the network is usually quite limited (not known, not measured), SQUAD automatically interpolates between the different network states to model signal propagation in the network. SQUAD assumes standard exponential functions but modifies the kinetic parameters of the function according to the network logic using concatenated exponential functions. The network is inevitably always a simplification regarding modulatory input from the remaining cell, as we model only about 30 proteins though the cell contains 5.000 proteins. To take this into account, the ground state (how strongly activated or inhibited at the start) was modified for several nodes in the simplified network according to available experimental data. This is given in Appendix A. For all other nodes, the ground state was set to zero. Next, the trajectories of full or partial activation down to no inhibition were calculated for the whole network and for all included proteins. However, only selected ones are shown in the figures to avoid cluttering the figures. Different mutational profiles and treatments were integrated into the dynamic simulation via the SQUAD perturbation function. Outcome predictions (Appendix A, “outcome”) were assessed by comparing to experimental readouts (proliferation and apoptosis in cell culture) as basic markers as well as considering all available co-mutations. The simulation initially describes only time units and activation strengths. Normalization of the activities and activation times was performed according to the collected experimental data: several iterative cycles between simulations, comparison with experimental data, and modifying the network topology and parameters accordingly. The resulting predictions are shown for the therapy-resistant cell line HCC44 and targeted combination therapy. All simulation protocols on parameters, simulated stimulation or inhibition, and stimulus time were saved in the SQUAD prt file format.

## 3. Results

### 3.1. Generation and Characterization of the 3D Lung Cancer Tissue Model

To generate a more reliable and in vivo-like preclinical tumor model, we developed a lung cancer model on a decellularized tissue matrix from porcine jejunum termed SISmuc. From one single pig, about 150 transwell cultures can be produced and used as inserts for 12-well or 24-well plates (Figure 1A).

After cell seeding, a homeostasis-like state is reached at about day 10 of 3D cell culture, and this state remains stable for at least one further week [13]. Dynamic culture conditions in bioreactors enhance cell growth (Figure 1B) and enable longer culture periods. A unique feature of our model is the preserved basement membrane enabling physiological anchorage of epithelial cells from which carcinomas derive. The addition of TGF-β1 induces EMT (increased vimentin staining, red in Figure 1D) and invasion across the basement membrane as a typical feature of carcinomas (arrows in Figure 1D). Cells grown in the tumor model are phenotypically characterized by established clinical markers. Regarding the expression of the lung adenocarcinoma markers thyroid transcription factor 1 (TTF1) and cytokeratin 7 (CK7), the adenocarcinoma-derived cell line HCC827 harboring an activating *EGFR* mutation well correlates with clinical adenocarcinoma specimens, whereas the *KRAS* mutated widely used cell line A549 isolated from an alveolar cell carcinoma is negative for both markers (Figure 1). However, the squamous cell carcinoma marker P63 (slightly positive in a single adenocarcinoma specimen) was strongly upregulated in both cell lines under 3D culture conditions (Appendix A). Further, we investigated the expression of the experimental but clinically relevant marker osteopontin, also known as secreted phosphoprotein 1 (SPP1), which was recognized as an indicator of tumor aggressiveness and metastatic potential and is discussed as a diagnostic biomarker. Immunohistochemically, both cell lines mentioned above were positive for SPP1 in 2D and 3D cultures (Figure 1J,M). Comparing the expression of other lung cancer-associated genes in the HCC827 cell line with preserved adenocarcinomatous phenotype under 2D and 3D conditions by PCR arrays revealed an upregulated transcription of anterior gradient protein 2 homolog (AGR2), carcinoembryonic antigen-related cell adhesion molecule 6 (CEACAM6), signal transducer and activator of transcription 2 (STAT2), transforming growth factor beta 1 (TGF-β1), TOX high mobility group box family member 3 (TOX3) and vascular endothelial growth factor A (VEGFA). All these genes were expressed at least four times higher in 3D than in 2D culture (Appendix A). Most of them are involved in carcinogenesis (AGR2 (connected to Mucin-1), STAT2 [24], TGF-β1), and cell adhesion (CEACAM6). Taken together, these results illustrate the strong impact of cell culture conditions on tumor cell growth, signaling, and biomarker expression and suggest a more cancer-like state of tumor cells in our 3D model in comparison to 2D culture.

### 3.2. Assessment of EMT, Differentiation, Stemness, and Invasion

For a more detailed biological characterization in the context of EMT (pan-cytokeratin (PCK), vimentin), we looked at differentiation (E-cadherin, Mucin-1) and stemness (CD44), and we quantified cell invasion after staining of the basal membrane component collagen IV. For this, two additional lung cancer cell lines were used besides HCC827 cells: H358 with a more epithelial phenotype and HCC44 with more mesenchymal features. Both cell lines harbor the *KRAS^G12C^* mutation, which can be targeted by the KRAS inhibitor ARS-1620. The two cell lines with more epithelial characteristics (HCC827, H358) were additionally treated with TGF-β1 to reveal general EMT-correlating patterns, as shown in Figure 2. In line with HCC827, H358 cells without TGF-β1 stimulation show a clear expression of the adherence junction protein and epithelial marker E-cadherin at cell-to-cell contacts, which is not present in HCC44 cells. Furthermore, both cell lines express the epithelial marker Mucin-1. Here, we observed in HCC827 cells a polarized apical location, in H358 cells a strong but basolateral expression, and the same expression pattern (but only weakly present) in HCC44 cells. In contrast to HCC827 cells, H358 cells in 3D culture (as well as HCC44 cells) show not only PCK but also vimentin expression. The stemness marker CD44 is highly expressed in HCC44 cells corresponding with the highest EMT status, but not in HCC827 cells and only weakly in H358 cells. Next, we stimulated the more epithelial HCC827 and H358 cells with TGF-β1 and observed a shift to a more mesenchymal phenotype showing E-cadherin- and Mucin-1-negative cells with stronger vimentin and CD44 expression. EMT status correlated with the expression of the stemness marker CD44. Independent of its location, Mucin-1 expression is inversely correlated with TGF-β1-dependent EMT induction (Figure 2).

TGF-β1 also led to the enhanced invasion of HCC827 and H358 cells. Interestingly, HCC44 cells were massively more invasive than HCC827 and H358 cells. We generated secondary resistant subclones named HCCresA2 and HCCresA3 by permanent EGFR inhibitor treatment of HCC827 cells with gefitinib (Iressa^®^, Absource Diagnostics GmbH, München, Germany). Drug resistance developed after about four months of treatment (Appendix A). These subclones exhibited increased invasiveness compared to their parental cell line (Figure 3). While EMT and invasiveness were interconnected in the tested cell lines, pathological assessment of the EMT markers cytokeratin and vimentin in 10 clinical samples of invasive lung adenocarcinomas revealed exclusive cytokeratin expression in the center of invasive tumor nodules; however, it does not represent the invasive front (Appendix A).

### 3.3. EMT Correlation with Drug Response

Next, we investigated how EMT correlates with drug response in our three cell lines carrying the clinically relevant *KRAS^G12C^* (H358, HCC44) and *EGFR* mutation (HCC827). Since tumors with *KRAS* mutations exhibit a high chemoresistance, termed primary resistance, we selected H358 and HCC44 cells as they both harbor the *KRAS^G12C^* mutation. This mutated protein is targetable with the allele-specific and covalent inhibitor ARS-1620, a derivative of which received accelerated approval by the FDA in May 2021. Our analysis shows a strong connection between EMT and drug response in both 2D and 3D cell cultures (Figure 4 and Appendix A). After low doses of ARS-1620, the more epithelial H358 cells show a reduction in cell number as well as in proliferation and dose-dependent apoptosis induction (Figure 4A–C). In the more mesenchymal and invasive HCC44 cells, no pronounced and dose-dependent effects on cell number, proliferation, and apoptosis could be observed (Figure 4A,B,D). Regarding the EGFR mutation, in the resistant subclones HCCresA2 and HCCresA3 permanently treated with the EGFR inhibitor, resistance correlates with EMT (PCK/VIM), loss of E-cadherin, downregulation of Mucin-1 and upregulation of CD44 (Figure 4E–G).

### 3.4. Set-Up of Combined In Vitro/In Silico Models with KRAS Signatures

To unravel the underlying dependencies of drug responses in this setting more systematically, we further applied an in silico model that displays a network map of signaling pathways (topology) for our *KRAS^G12C^* models. This topology is given in a machine-readable format (using CellDesigner [15]). It can be used to simulate systemic drug responses in specific mutational backgrounds by the application of the SQUAD software. To achieve a more clinically relevant model, we integrated into the topology common co-mutations of *KRAS* found in over 1000 clinical lung adenocarcinoma patients [10], most frequently being *P53* (about 40%) and *LKB1/STK11* (about 20%), and *KEAP**1*, to enable the efficient patient-specific translation of the model from bench to bedside (Figure 5A). The resulting model topology of key pathways and interactions in lung adenocarcinoma is given as the Appendix A.

However, such a lung cancer network is always a simplification, and we wanted to include the effect of input from outside of the topology and consider differences between the H358 and HCC44 cell lines. Hence, to take these activities of important nodes into account that specifically acted differently in H358 and HCC44 cells looking at untreated and treated with the KRAS inhibitor ARS-1620, we pre-set ground state activities in our model estimated from the literature and experimental data (Appendix A, gray columns). In preparation for individual patient predictions, mutational differences of the cell line H358 (responder to the KRAS inhibitor ARS-1620) and HCC44 (non-responder) were adjusted to certain activation levels in order to stratify subgroups that could be relevant for drug efficacy. In detail, H358 cells display a *TP53* and a *PIK3CG* mutation together with *KRAS*, but the *LKB1* gene is wildtype [25]. On the other side, HCC44 cells harbor an *LKB1*-, *KEAP1*-, and *SMARCA4* mutation in addition to *P53* [25,26], and a higher expression of c-MYC is reported [27]. Further differences between these two cell lines are a lower EMT status, stemness (Figure 2), invasion (Figure 3), and proliferation (Figure 4) in H358 compared to HCC44 cells. For successful simulation, according to the experimental data, we had to add (next to EMT activation by vimentin) an activation link in the network topology from vimentin to EKR2 that, in turn, inhibits proliferation. One important node for therapeutic success in the network is GSK-3β (Appendix A).

In our simulation, a change in the state of the node shows a significant change in the apoptotic response for H358 cells (Appendix A). In our simulation, the co-mutation *KEAP1*, which shows a loss of function in HCC44, is indicated to be an important regulator in the apoptotic and proliferation pathway by hindering therapeutic responses. In contrast, *P53* and other common co-mutations have similar effects on simulations in both cell lines. After these adaptations of the starting ground state (Appendix A), the experimentally measured drug responses in vitro toward ARS-1620 regarding proliferation, apoptosis, and EMT in H358 and HCC44 toward ARS-1620 treatment could be correctly simulated with the software SQUAD (Appendix A, simulation output). Important nodes were graphically represented regarding strength (*y*-axis) over time (*x*-axis) (Figure 5B).

### 3.5. TGF-β1-Induced EMT Does Not Mediate Resistance toward Targeted Therapies

The correlation between EMT and drug resistance is widely shown in the literature and in our previous experiments (Figure 4). Next, we wanted to investigate whether EMT is a marker or a marker of resistance. As TGF-β1 is a strong inductor of EMT, we investigated changes in the drug response of the more epithelial H358 and HCC827 cells upon stimulation with this growth factor.

As described above, the stimulation of H358 and HCC827 cells with TGF-β1 resulted in a progressed EMT phenotype, pointed out by a lower expression of epithelial markers and a stronger expression of mesenchymal markers (Figure 2 and Figure 6A,C). Both H&E and immunofluorescence stainings indicated reduced cell numbers in H358 and HCC827 tumor models after treatment with either ARS-1620 or gefitinib, independent of TGF-β1 (Figure 6A,C). While treatment with TGF-β1 alone already resulted in reduced proliferation indices for both cell lines compared to the untreated controls, additional treatment of the H358 and HCC827 cells with ARS-1620 or gefitinib, respectively, led to an even stronger decline in the proliferation rate (Figure 6B,D). The increase in apoptosis in cultures treated with both TGF-β1 and ARS-1620 or gefitinib was comparable to that in cultures solely treated with the corresponding targeted therapy (Figure 6B,D). In summary, the tissue architecture, the reduced proliferation indices, as well as the increase in apoptosis provide evidence that EMT induction with TGF-β1 alone is not sufficient to mediate resistance of H358 and HCC827 cells toward either ARS-1620 or gefitinib.

### 3.6. EMT Status and CD44 Expression Are No Predictors of Drug Response in PDX Cell Lines

To investigate the correlation between EMT and stemness on a larger scale, we used four different *KRAS^G12C^*-mutated patient-derived xenograft (PDX) lung cancer cell lines (LXFA 983, LXFL 1072, LXFL 1674, LXFA 2184), which were kindly provided by our collaborator Oncotest (Charles River, Freiburg). Interestingly, no clear interdependency between EMT phenotype and sensitivity toward the KRAS^G12C^ inhibitor ARS-1620 was observed in 2D or 3D cell culture (Figure 7). The most epithelial cell line LXFA 983 exhibited the highest IC_50_ value toward ARS-1620 in 2D (IC_50_ = 14.6 µM) and did not display sensitivity toward the inhibitor in 3D (viability about 95%). However, LXFL 1072 cells differentiated into a more epithelial phenotype when cultured in 3D. Remarkably, they switched from a resistant state in 2D (IC_50_ = 7.6 µM) to drug response in 3D (viability 65%) comparable to LXFL 1647 cells (viability 70%), which are clearly the most sensitive PDX cells in 2D (IC_50_ = 1.1 µM) but displayed a high grade of EMT in 2D and 3D conditions (Figure 7B,C).

In the four PDX cell lines, we observe a black and white pattern regarding E-cadherin and CD44 staining (Figure 7A), which underlines a connection between a high EMT status and stemness in LXFL 1674 and LXFA 2184, but this does not correlate with drug resistance (Figure 7C).

### 3.7. Combination Strategies to Overcome Resistance in HCC44 Tumor Models

Due to these inconsistent results for EMT status regarding resistance, we tested how we could break higher resistance in *KRAS^G12C^*-mutated HCC44 cells compared to H358 cells by targeting other possible resistance determinants. Homeostasis is the equilibrium of the cell. In a healthy cell, the differentiation pathways reliably help the cell to carry out its specific functions. In lung cancer, the system state that is actively preserved in the cancer cell is high proliferation, low differentiation, and low apoptosis. A combinatorial treatment allows rectifying this system state by combining drugs targeting specific pathways, for instance, a kinase inhibitor blocking proliferation with another drug stimulating apoptosis.

Therefore, we performed several combination tests together with the KRAS^G12C^ inhibitor ARS-1620 suggested by either genome-scale CRISPR interference (CRISPRi) studies, which identified essential genes when KRAS as a tumor driver is inhibited [28] or by described resistance mechanisms as responses to KRAS^G12C^ inhibition [29,30]. These combination strategies included metformin as an AMP-activated protein kinase (AMPK) activator as well as receptor tyrosine kinase inhibitors (RTK-Is) such as erdafitinib and crizotinib targeting the fibroblast growth factor receptor (FGFR) or anaplastic lymphoma kinase (ALK), respectively. In addition, SHP099 inhibiting the Src homology region 2 domain-containing phosphatase 2 (SHP2) was assessed, and to further hinder feedback loop activation, one triple therapy with gefitinib, SHP099, and ARS-1620 was tested. The combined use of an SHP2 inhibitor and RTK-Is in *KRAS* mutant NSCLC cells was promising by suppressing stemness in vitro [31]. While we could find crizotinib as a promising drug candidate with a low IC_50_ value for HCC44 cells in 2D, we could not find any enhanced effect when this therapy was combined with ARS-1620 (Appendix A). In 3D, MTT assays revealed that none of the monotherapies and also none of the combinatorial treatments with ARS-1620 could reduce the viability of HCC44 cells to less than 75% of the untreated and that metformin even resulted in increased viability (Figure 8A). In former publications, drug efficacy in our 3D models better correlated with clinical results than that in 2D and animal experiments [15,17]. Here, we again observe a difference in drug response between 2D and 3D cultures, demonstrating the severe impact of culture conditions. As we assume that our 3D model delivers improved preclinical predictions, we used 3D results of non-effective treatments to optimize our existing in silico network topology [15], which in the first place predicted effective drug responses in in silico simulations similar to 2D culture testing. Methodic details of how we achieved this optimization can be found in the Appendix A.

Looking for further determinants of resistance, we came across the oncogenic cooperation between KRAS and c-MYC, driving invasion in KRAS^G12D^ mouse models [32]. This interaction potentially had an impact on our 3D test system’s sensitivity toward ARS-1620. Databases indicate that there is a higher expression of c-MYC in HCC44 cells than in H358 cells [33,34], which we could confirm on protein level in our 3D tumor model (Appendix A). After treatment with inhibitors of KRAS^G12D^, AURKA mediates its reactivation [30]. Treatment with an AURKA inhibitor also leads to the degradation of c-MYC [35]. Hence, we tested the AURKA inhibitor alisertib as a promising candidate for sensitizing the HCC44 cells to treatment with ARS-1620. In 2D culture, the IC_50_ value of alisertib lay in the sub-micromolar range, with and without ARS-1620. Treatment of human primary fibroblasts from healthy tissue with alisertib and ARS-1620, even at high concentrations, merely resulted in minor growth inhibition, indicating cytocompatibility (Appendix A). In our 3D tumor model, the combination of ARS-1620 with alisertib proved to be most effective in reducing viability (Figure 8A). This finding was underpinned by a significantly reduced cell number and proliferation index (Figure 8B,C). In line with these results, the optimized in silico simulation predicted an enhanced response of the HCC44 cell line to the aforementioned combination therapy, thereby independently supporting the findings of the in vitro experiments.

The in silico adjustments according to the experimental results for individually targeted therapies with crizotinib, SHP099, gefitinib, and alisertib at the start for the ground state are given in Appendix A. Simulation results for readout parameters (in particular apoptosis and proliferation) are given in Appendix A. The software could calculate the results for any combination therapy of choice. We see that some of the nodes change in comparison to the monotherapy within the ground state (blue color) (Figure 8D). Combination therapy simulations of SHP099 and gefitinib can be found in the Appendix A.

Furthermore, the correlation between the sensitizing effect of alisertib and c-MYC expression was supported by the treatment responses of the four PDX cell lines harboring the *KRAS^G12C^* mutation. The cell lines LXFL 1072 and LXFL 1674 with the highest c-MYC expression displayed the most pronounced reduction in viability. Vice versa, the LXFA 983 cell line with the lowest c-MYC expression was the most therapy-resistant (Appendix A).

## 4. Discussion

In this study, we investigated EMT, stemness, invasion, and drug resistance in several lung tumor cell lines and PDX-derived cell lines grown on a tissue matrix with the support of an in silico model. Several conclusions could be drawn: (i) EMT is more a marker than a maker of resistance, (ii) tissue context has an impact on EMT status, gene expression, and drug response, (iii) intrinsic factors are more important to invasion than EMT as confirmed by clinical specimens, and (iv) EMT correlates with stemness. Our results suggest that EMT is overestimated as a determinant of invasion and resistance, at least in lung cancer. We integrated frequent co-mutations of *KRAS* into cell line-specific in silico models to unravel complex interdependencies and define patient subgroups for clinical studies. As proof of concept, we demonstrated concordance between in vitro testing and in silico simulation of combinatorial treatments of the *KRAS*-mutated HCC44 cell line, further displaying high c-MYC expression and harboring *LKB1*, *P53*, and *KEAP1* co-mutations. Hence, this work mimics individual patient tumor signatures for later clinical application.

### 4.1. D Tissue Models for More Realistic Preclinical Testing

Models better representing tumor biological aspects are demanded by clinicians [36]. Different 3D tumor models are currently used in the field of cancer research, including spheroids and organoids. However, we see specific advantages of our biological scaffold SISmuc regarding this article’s issues as it allows for cell-ECM adhesion and the study of EMT processes and invasion due to apical-basal polarity of SISmuc-based models and a preserved basement membrane. We succeeded in generating different surrogate models for a variety of tumor entities, including breast, colorectal, and lung cancer [14,37,38]. Almost all commercially available cell lines tested so far are attached to the biological scaffold, and stable tumor models could be subsequently generated within 14 days. SISmuc tumor models are a versatile tool, also permitting co-cultures of tumor cells with fibroblasts or immune cells [13,39]. By successfully serving as a substitute for animal experiments in the testing of CAR T-cells, the SISmuc demonstrated its close correlation with in vivo conditions [39,40]. Still, the decellularized jejunum is not organotypic for the mentioned tumors, and ECM components highly differ between different organs (reviewed in the work of [41]), which is why characteristics of cancer cells might be differentially influenced. Additionally, the decellularization of the tissue is a time-consuming and labor-intensive process.

Previous work has shown the importance of cues from the ECM and especially basement membrane proteins for the cellular phenotype of both normal and cancer cells [42,43,44,45]. Therefore, we investigated the expression of clinically established lung carcinoma markers in our lung tumor models [46]. The HCC827 cell line shows homogeneous staining for the lung adenocarcinoma markers TTF1 and CK7 in both 2D and 3D cultures, reflecting the maintenance of the adenocarcinomatous phenotype. Similarly, the A549 cell line’s expression pattern of the two markers was widely stable across culture conditions. However, these cells are negative for TTF1, and staining for CK7 is markedly weaker, especially in the 3D culture. Intriguingly, both cell lines exhibit the squamous cell carcinoma marker P63 in the 3D culture while they do not in the 2D culture. Reasons for P63 expression in tumors derived from P63-negative tissues might be a redifferentiation toward a squamous phenotype or the acquisition of stem cell properties [47,48]. We could previously show a higher expression of stem cell markers in cells cultured on the matrix SISmuc in colorectal cancer models [13,47].

In addition to the immunohistochemical markers relevant to clinical subtyping, we investigated the expression of SPP1 (also known as osteopontin). In various cancer types, including lung, breast, colon, and prostate cancer, overexpression of SPP1 is associated with tumor invasion, metastasis, and poor clinical outcome [49,50,51,52]. For NSCLC, SPP1 has been proposed as a biomarker used for diagnosis and surveillance [53,54]. Both cell lines, A549 and HCC827, showed largely homogeneous staining for SPP1 in 2D and 3D cultures, comparable to that of lung adenocarcinoma specimens. These preliminary data suggest SPP1 signaling in our model, which might promote its invasive and metastatic capacity [55,56,57], supporting its eligibility for our studies. In a more large-scale analysis, PCR array results showed the upregulation of several genes related to cancer progression, indicating that 3D culture better reflects the conditions found in in vivo carcinomas. CEACAM6, being important for cell adhesion during cancer progression [58], was detected to be upregulated in 3D culture, stressing the importance of growth conditions tumor cells are exposed to.

### 4.2. EMT Correlation to Drug Resistance and Invasion

EMT and MET are fundamental processes during embryonal development and are discussed as key players in cancer [59,60,61,62]. However, it became evident that cancer cells do not necessarily pass through an EMT to spread to distant organs, and this process is highly context-dependent [63,64]. In line with this, we observe in our study a far higher invasion by one specific cell line (HCC44) than in other cell lines with a similar EMT status induced by TGF-β1, suggesting other intrinsic factors to be more important to driving invasion. Consistency of these findings with the clinic is shown by the fact that in samples of 10 lung adenocarcinoma specimens derived from the tumor center, however, all invasive tumor cells strongly expressed PCK but not vimentin. Next to PCK and vimentin, we investigated E-cadherin in our 3D models as a further marker of epithelial differentiation. This adherence junction protein is connected to β-catenin, which is observed to translocate to the nucleus for EMT induction at the invasive front of CRC [65]. The investigation of β-catenin localization is part of the clinical routine in CRC but not in NSCLC.

A correlation or even a dependency of resistance on EMT is claimed in several publications [1,66,67]. Indeed, we could find a higher EMT status in secondary resistant cells (HCCres cells) after permanent gefitinib treatment, in addition to higher resistance in the more mesenchymal of two KRAS mutant cell lines (H358, HCC44). While TGF-β emerged as a central player in cancer drug resistance about 10 years ago [2], we could conversely not observe resistance in HCC827 and H358 cells after EMT induction by TGF-β. Interfering with TGF-β signaling has been a great hope in cancer therapy for a decade with limited success [68,69]. Finally, in PDX cell lines with four different degrees of EMT status, the most epithelial one surprisingly exhibited the highest resistance together with the most mesenchymal cell line. Taken together, these observations suggest EMT is neither necessary nor sufficient for drug resistance development. Notably, reversion of EMT must be handled carefully in the clinic as MET could promote metastasis formation at distal sites [4].

EMT is also discussed to be correlated with stemness [1,70]. There is some evidence of CD44 being a cancer stem cell marker [71]. CD44 is a significant and clinically relevant prognostic marker in NSCLC patients [72]. In combined in vitro and mathematical feedback-loop studies, CD44 was also confirmed as an important factor in maintaining an EMT hybrid status [73]. In line with this, we observed in all experiments a co-regulation of EMT and CD44 but no predictive value for drug resistance. The two most sensitive PDX cell lines were in between the most epithelial and mesenchymal cells with their EMT phenotype. Additionally, while one of these two cell lines expressed CD44, the other did not. We further observed the repression of transmembrane glycoprotein Mucin-1 in all cells displaying a progressed EMT phenotype in the 3D tumor model.

### 4.3. An In Vivo/In Silico Platform for Testing Targeted Therapies

In order to generate a platform suitable for biomarker-signature-based testing, we connected our 3D tissue tumor model with an in silico model [13,15,17]. To adapt the model for specific tumor cell lines, common co-mutations in the large subgroup of *KRAS*-mutated NSCLC were also integrated, which correspond to patients’ tumor mutations observed in the clinic [10]. For optimization of the network, we used experimentally determined parameters of the two cell lines H358 and HCC44 as a responder and non-responder to the KRAS inhibitor ARS-1620, respectively. To find hints that higher resistance in HCC44 cells is not only based on multi-drug resistance (MDR) proteins, we checked the expression of MDR and ABC transporters in an available online database [74,75] and did not find any expression. The in silico model integrates “EMT” as a systemic output similar to “apoptosis” and “proliferation”. Signaling, which leads to EMT, is interconnected to several cascades. By this, changes in EMT levels can be observed, but the model does not support EMT as a marker or maker for drug resistance and serves primarily as an effective screening tool for combinatorial drugs. EMT is discussed to be a marker of KRAS independency [76,77]. In the case of KRAS independence, co-mutation signatures gain importance for combinatorial treatment. To integrate EMT in simulations correctly, the cell line HCC44, as a non-responder to the KRAS inhibitor, had to show a higher EMT than the responder cell line H358, as shown in Figure 5. Combinatorial treatments were then tested in HCC44 with high EMT status.

The combination of RTK inhibitors together with KRAS inhibition is assumed to have combinatorial or even synergistic drug effects in the corresponding cancer cells. (Reviewed in [78]). There is further evidence that the inhibition of SHP2 can inhibit potentially negative feedback loops in KRAS signaling in a more general way and thereby increases the efficacy of KRAS inhibition in different *KRAS^G12C^*-mutated cell lines [79]. Here, we could always observe a slightly increased effect when RTK inhibitors were combined with ARS-1620 on the 3D tumor models. Still, the effect of these combinations was marginal and not sufficient to overcome the primary higher resistance of HCC44 cells. Next, we met the challenge to simulate in silico the higher resistance of HCC44 to combinatorial treatments observed in 3D culture. Based on the fitting of HCC44 cells under treatment with ARS-1620 (Appendix A), control simulations with three combinatorial targeted treatments (SHP2-inhibitor, crizotinib, gefitinib) together with the KRAS inhibitor were performed to optimize network connections in the signaling topology until in silico only marginal effects could be observed as it can be seen in the clinic. For focused optimization, we used the obvious difference between 2D and 3D culture: in 3D culture the proliferation and thereby also the metabolism is lower compared to 2D conditions. Since EMT is also regulated in the context of metabolism, we linked the mitosis regulator AURKA to EMT, c-MYC [80,81], and LKB1 [82] in the in silico topology. This enables us to simulate also the non-effective AMPK activation by metformin. To identify an effective combinatorial therapy, we looked for other possible vulnerabilities in HCC44. As HCC44 cells show a *P53* mutation and a high c-MYC expression [27] which is also maintained under 3D conditions (Appendix A), we tested the AURKA inhibitor alisertib, which is reported to induce c-MYC degradation in P53 mutant cancer [83]. This inhibitor has been tested in clinical trials for multiple cancers and is the only one that reached phase III (Reviewed in [80,82,84]). The relevance of AURKA in lung cancer patients is stressed by the correlation of poor outcomes with AURKA expression, which could also be correlated to resistance toward the KRAS^G12C^ inhibitor in vitro [85]. Here, we could demonstrate that this inhibitor was more effective than the KRAS inhibitor alone (90% viability), whether used alone (75% viability) or together with the KRAS inhibitor ARS-1620 (60% viability). The benefit of this combinatorial treatment in NSCLC showing *KRAS* mutation and high c-MYC expression was supported by the differential efficacy in four PDX cell lines decreasing the viability to up to 50%. The two most sensitive PDX cell lines display both a *KEAP-1* co-mutation but only one single cell line, a *P53* mutation suggesting the *KEAP-1* mutation to be more relevant for drug efficacy in this context. Furthermore, they display a completely different EMT status, which argues against EMT being a conclusive marker of KRAS independency. We are aware that this needs further investigation. The simulations of the combination therapy with ARS-1620 and alisertib could be predicted based on the network and were made independently from the experimentally determined results (Figure 8). Interestingly, KEAP-1 is one important ground state parameter of higher resistance in HCC44 cells compared to H358 cells (Appendix A). Thereby, our platform is now available for preclinical application.

In general, in silico modeling allows us to have fewer experiments by pre-testing various drugs and their combinations. We can achieve cell line- and thereby biomarker-specific in silico models. For clinical transfer, we would simply match the mutation profile from tumor biomarker analysis with optimal therapy strategies explored before in our in vitro/in silico platforms using insights from network modeling [86]. Our combined in silico/in vitro results are proof of concept that a systematic understanding of the tumor cell-specific network and its different pathway dependencies on the biomarker signature are the key to a rational combinatorial targeted treatment. In contrast, individual markers such as EMT do not correlate sufficiently well with the cellular systems response and treatment success.

Though we focus our analysis on the treatment of lung cancer and, in particular, on NSCLC, our in vitro/in silico platform can be easily adapted to other cancer entities. We currently investigate this in breast cancer [13] and colon cancer [14]. The pathway-centric view of our platform is also very useful to reveal proliferative and apoptotic pathways and immune-modulator and immune-suppressive interactions between tumor and immune cells, along with implied cancer engines. A cooperative effect of *KRAS* mutation and the expression of c-MYC for invasion, as also observed in HCC44 cells in our model, and for immune modulation has been shown previously in animal studies for lung and pancreatic cancer [32,87]. We are aware that large-scale phospho-proteomic and expression data would add important value to our model. This and the comparison with patient data will be one focus of further studies. As a starting point, we also analyzed established clinical markers in our models and vice versa, EMT status and cell invasion in clinical specimens. Furthermore, we integrated common co-mutations of *KRAS* found in NSCLC patient cohorts into the in silico network topology.

## 5. Conclusions

In this study, we investigated different functions of EMT related to the microenvironment of tumor cells, integrating parts from the biological context of carcinomas in a tissue tumor model for lung cancer. Of note, we think that the tumor biological dogma of EMT being the central factor for invasion and drug resistance should be put into perspective. This is supported by clinical observations, especially for lung cancer. For the reduction in the still high attrition rates of preclinical models, a paradigm shift is mandatory to models that could reflect characteristics of the tissue context of tumor cells. For the large and challenging subgroup of *KRAS*-mutated NSCLC patients, an in silico network was used for a systematic analysis of combinatorial effects, also considering common co-mutations found in the clinic. The pathways and effective targeting options and combinations revealed by our in vitro/in silico platform help to pave the road for patient stratification. This will involve clinical studies according to sequencing data in *KRAS*-mutated NSCLC patients. This strategy has not to be restricted to targeted therapies but should also involve strategies to overcome immune-suppressive hurdles in the tumor microenvironment.

## Figures and Tables

**Figure 1 cancers-14-02176-f001:**
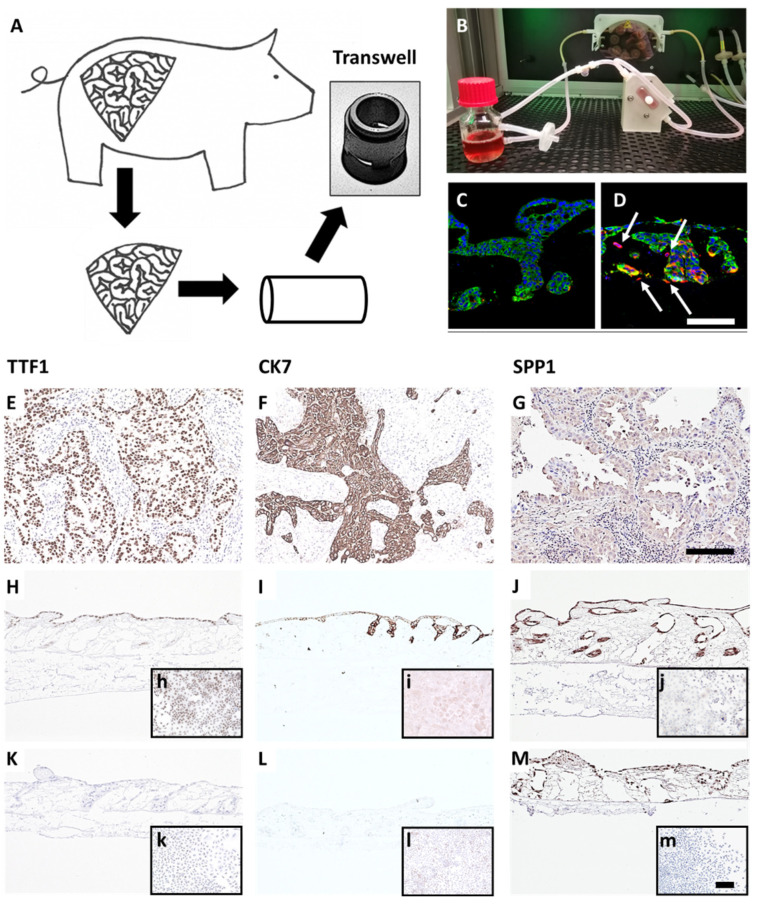
A preclinical tissue tumor model reflecting EMT, invasion, clinical and biological markers. (**A**) 3D tumor model based on porcine jejunum. (**B**) Dynamic cell culture in a flow bioreactor. (**C**) HCC827 lung cancer cells on SISmuc (Pan-cytokeratin, green) and (**D**) HCC827 cells on SISmuc + TGF-β1 in a flow bioreactor, arrows: vimentin-positive cells (red) invading across the basement membrane the collagen matrix. DAB staining of TTF1, CK7, and SPP1 on paraffin sections: adenocarcinoma of the lung (**E**–**G**), HCC827 cells on SISmuc under static culture conditions (**H**–**J**), inserts: HCC827 in 2D, A549 cells on SISmuc under static culture conditions (**K**–**M**), inserts: A549 in 2D. Scale bar in (**D**,**G**) = 100 µm for (**C**–**M**); scale bar in m = 100 µm for (**h**–**m**).

**Figure 2 cancers-14-02176-f002:**
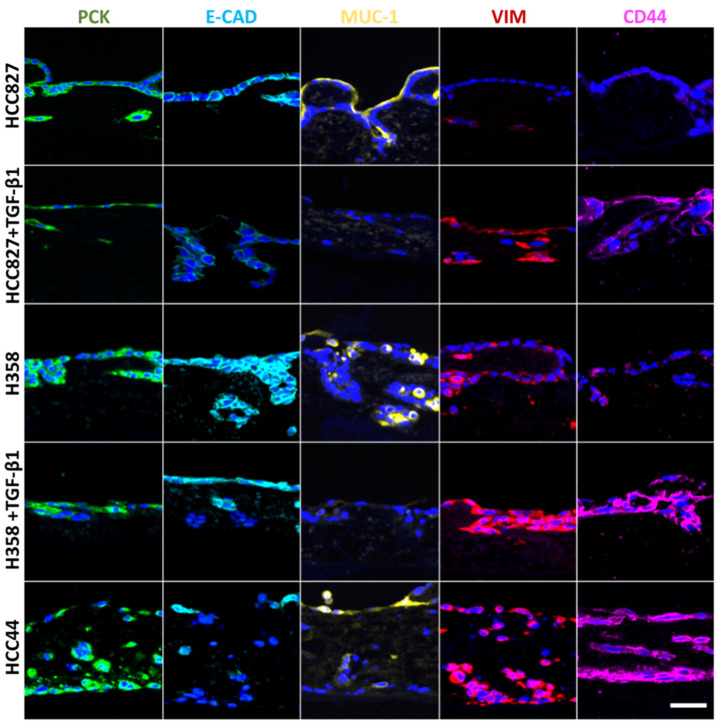
EMT correlates with stemness, is inducible by TGF-β1, and is inversely correlated with Mucin-1 expression, independent of its location. Three different 3D tumor models (HCC827, H358, HCC44) with and without TGF-β1 (2 ng/mL) treatment are stained for different markers of EMT and stemness: pan-cytokeratin (PCK, green), E-cadherin (E-CAD, light blue), Mucin-1 (MUC-1, yellow), vimentin (VIM, red), and CD44 (purple). Cell nuclei are counterstained with DAPI (blue). Scale bar = 50 µm for all images.

**Figure 3 cancers-14-02176-f003:**
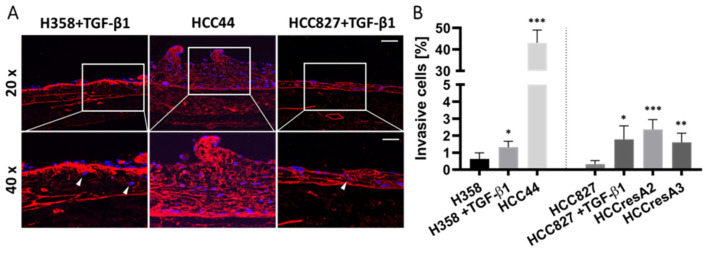
EMT correlates to some extent with invasion, but the intrinsic invasion of HCC44 exceeds TGF-β1-induced invasion in H358 and HCC827. (**A**) Collagen IV (red) immunofluorescence staining with DAPI (blue) counterstaining of HCC44 tumor models or H358 and HCC827 3D models treated with 2 ng/mL TGF-β1. Invasive cells are indicated with white arrowheads. HCC827 gefitinib-resistant subclones A2 and A3 display a similar degree of invasion as parental HCC827 stimulated with TGF-β1. Scale bar = 100 µm (upper panel); 50 µm (lower panel); *n* = 4. (**B**) Quantitative evaluation of invasive cells; *n* = 4. Significance determined with unpaired t-tests versus H358 or HCC827, respectively. *: *p* ≤ 0.05, **: *p* ≤ 0.01, ***: *p* ≤ 0.001.

**Figure 4 cancers-14-02176-f004:**
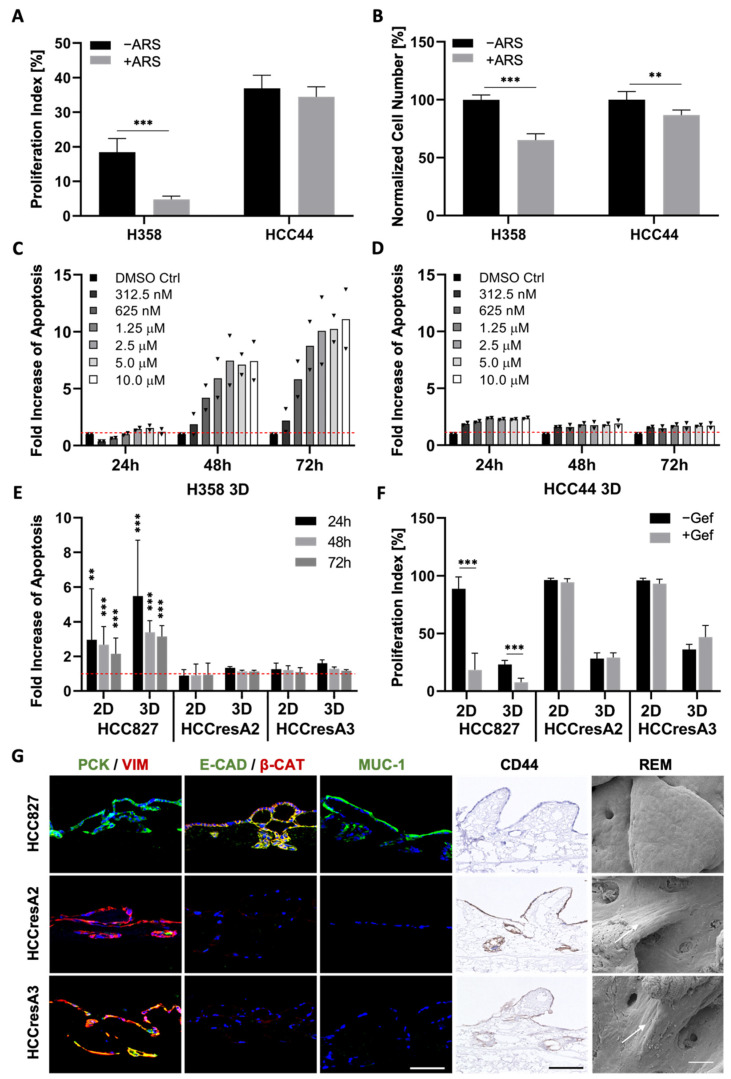
EMT, dedifferentiation, and stemness correlate with primary as well as secondary resistance. (**A**) Proliferation indices and (**B**) normalized cell numbers of H358 and HCC44 cells on 3D SISmuc tumor models with or without 1 µM ARS-1620 treatment; *n* = 4. Fold increase in apoptosis over untreated control evaluated by M30 ELISAs of (**C**) H358 and (**D**) HCC44 cells in 3D after the treatment with indicated concentrations of ARS-1620. Red line indicates the baseline apoptosis of the DMSO control. Triangles (▼) represent values from single biological replicates; *n* = 2. (**E**) Fold increase in apoptosis over untreated control (red line) and (**F**) proliferation indices in HCC827, HCCresA2 and HCCresA3 cells in 2D and 3D after treatment with 1 µM gefitinib; 4 ≤ *n* ≤ 13. (**G**) Immunohistochemistry staining of CD44 and immunofluorescence staining of EMT markers pan-cytokeratin (PCK, green), vimentin (VIM, red), E-cadherin (E-CAD, green), β-catenin (β-CAT, red), and Mucin-1 (MUC-1, green) of HCC827, HCCresA2, and HCCresA3; scale bar = 100 µm. Reflection electron microscopy (REM): white arrows indicate the elongated shape of resistant cells; scale bar = 50 µm. Significance determined with unpaired t-tests. **: *p* ≤ 0.01, ***: *p* ≤ 0.001.

**Figure 5 cancers-14-02176-f005:**
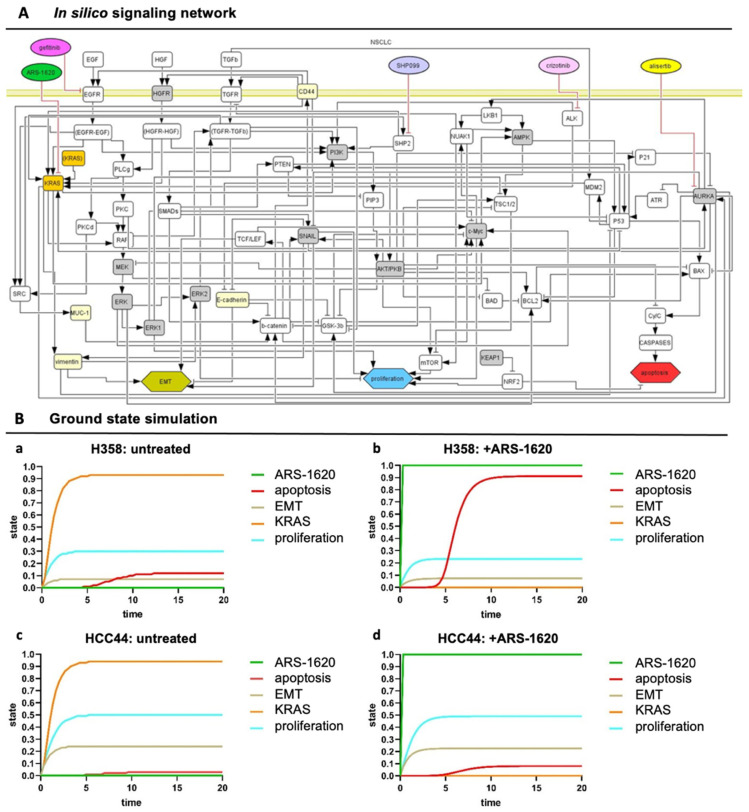
In silico signaling network topology used for H358 and HCC44 therapy simulations. (**A**) The network of protein–protein interactions (rounded rectangles) and cellular responses (hexagons) integrates common co-mutations of *KRAS*. Interactions: arrows activating interactions, blunted arrows inhibitory interactions; white rectangles: interacting nodes, gray rectangles: assumed to be constant, yellow rectangles: constant nodes tested experimentally, orange rectangles: KRASmt node. (All coding also applies to the following simulations) (**B**) SQUAD calculates the activity changes and responses for every node in the network in detail. H358 and HCC44 treated with ARS-1620, with H358 being a responder to KRAS inhibition seen by induction of apoptosis and slight reduction in proliferation, which is not the case in simulations for HCC44 cells. (**a**) Untreated H358, (**b**) H358 treated with ARS-1620, (**c**) untreated HCC44, and (**d**) HCC44 treated with ARS-1620. Only the readout of interesting nodes of our network is shown. However, all nodes of the network as given in (**A**) are simulated and available in their trajectories so that novel drugs, as well as the detailed response of the whole network, can be studied.

**Figure 6 cancers-14-02176-f006:**
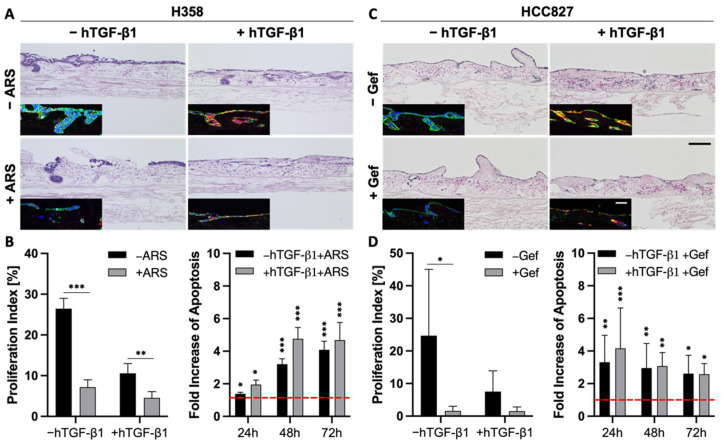
EMT is more a marker than a maker of resistance. (**A**,**C**) H&E and immunofluorescence stainings of pan-cytokeratin (green) and vimentin (red) of 3D tumor models with H358 (**A**) and HCC827 cells (**C**) treated with 2 ng/mL TGF-β1 and 1 µM ARS-1620 or 1 µM gefitinib, respectively; *n* = 4. Scale bars = 100 µm. Proliferation indices and fold increase in apoptosis over untreated control of H358 (**B**) and HCC827 (**D**) cells in 3D after the treatment with 2 ng/mL TGF-β1 and 1 µM ARS-1620 (H358) or 1 µM gefitinib (HCC827). Red line indicates the baseline apoptosis of the corresponding controls. Significance determined with unpaired t-tests. *: *p* ≤ 0.05, **: *p* ≤ 0.01, ***: *p* ≤ 0.001; 4 ≤ *n* ≤ 6.

**Figure 7 cancers-14-02176-f007:**
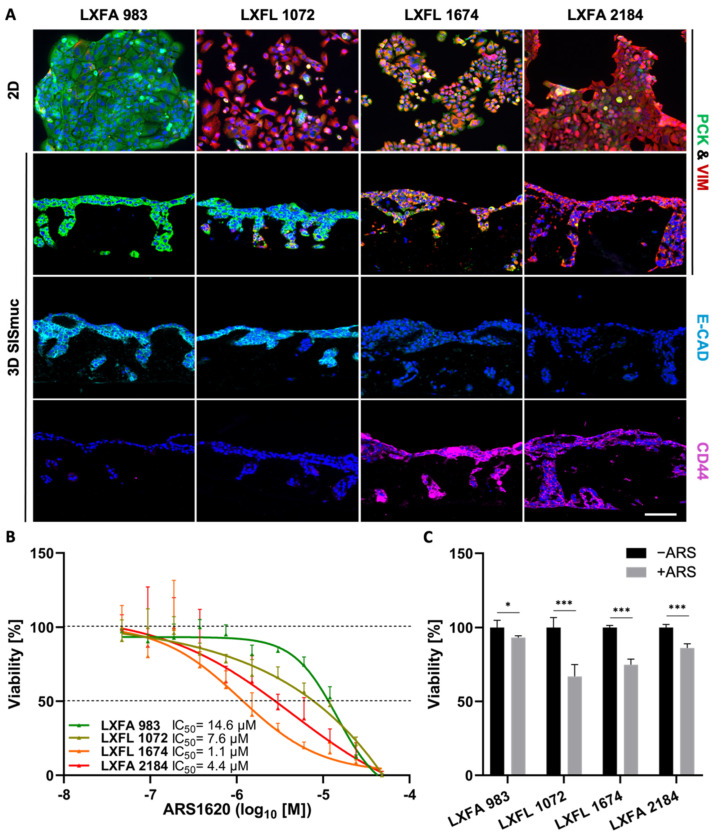
Neither EMT status nor CD44 expression correlates in PDX cell lines with drug response in 2D and 3D. (**A**) Pan-cytokeratin (PCK, green), vimentin (VIM, red), E-cadherin (light blue), and CD44 (purple) immunofluorescence staining of *KRAS^G12C^*-mutated PDX-derived lung cancer cells in 2D (*n* = 2) and 3D (*n* = 2). Scale bar = 100 µm. (**B**) CellTiter-Glo viability assay of PDX-derived lung cancer cells after treatment with increasing concentrations of ARS-1620. Calculated IC_50_ values for 2D cultures are indicated. The picture shows the IC_50_ curve of one representative experiment of two independent assays; *n* = 2. (**C**) MTT-assay of 3D SISmuc tumor models seeded with the PDX-derived cell lines and treated with 1 µM ARS-1620 for 72 h. Significance determined with unpaired t-tests. *: *p* ≤ 0.05, ***: *p* ≤ 0.001; *n* = 4.

**Figure 8 cancers-14-02176-f008:**
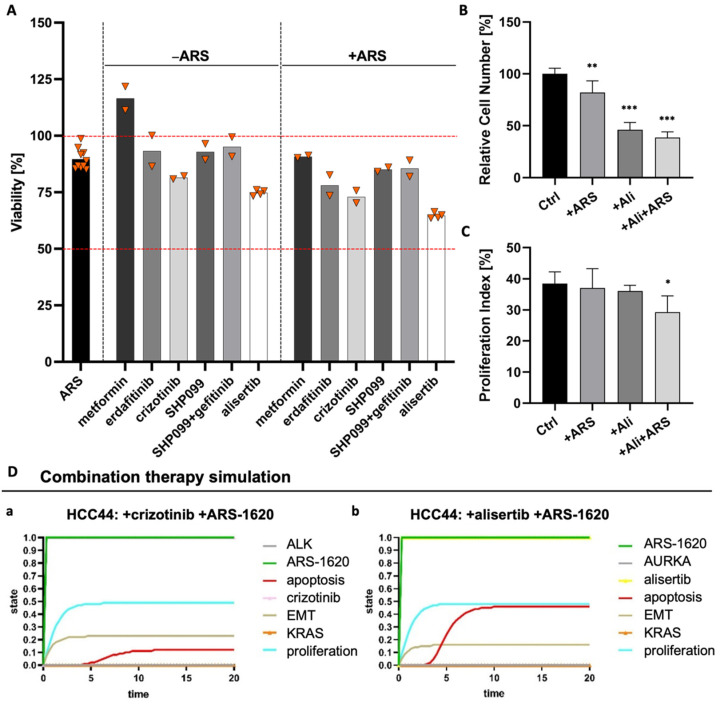
Combination of ARS1620 with AURKA inhibitor alisertib as the most effective combination in 3D HCC44 tumor models. (**A**) MTT assays of HCC44 tumor models treated with 1 mM metformin or 5 µM SHP099, erdafitinib, gefitinib, crizotinib, or alisertib. Drugs were tested either in monotherapies or in combination with 1 µM ARS-1620. Triangles (▼) represent values from single biological replicates; *n* ≥ 2. (**B**) Relative cell numbers and (**C**) proliferation indices of HCC44 cells in 3D after the treatment for 72 h with 1 µM ARS-1620, 5 µM alisertib and the combination of both inhibitors; *n* = 4. (**D**) In silico combination therapy simulations of (**a**) HCC44 treated with crizotinib and ARS-1620 and (**b**) HCC44 treated with alisertib and ARS-1620. Color code for different readout parameters is given on the right side of the figure. Significance determined with unpaired t-tests. *: *p* ≤ 0.05, **: *p* ≤ 0.01, ***: *p* ≤ 0.001.

## Data Availability

All data for this publication are included in the manuscript and its Appendix A. The software used (SQUAD and CellDesigner) are publicly available as indicated in the manuscript, and the model files used (H358, HCC44) are given as Appendix A.

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
