# Peer review of "EMT, Stemness, and Drug Resistance in Biological Context: A 3D Tumor Tissue/In Silico Platform for Analysis of Combinatorial Treatment in NSCLC with Aggressive KRAS-Biomarker Signatures"

_cancers, 2022, doi:10.3390/cancers14092176_

Round 1

Reviewer 1 Report

This manuscript described that EMT is not a significant marker or predictor of drug resistance in contrast to the previous notions, from the studies with KRAS inhibitors on 3D model of lung cancer cells. In addition, authors analyzed an in silico model, and identified an aurora kinase A inhibitor as the most promising combinational drug with a KRAS inhibitor.

Although I agree that the significance of EMT for drug resistance is still in debate, the evidences for the authors’ conclusion in the manuscript are not well presented. The results should be rearranged more consistently according to their hypothesis or argument.

The significance of the authors’ conclusion from an in silico model is not well described. Is it supporting that EMT is not a significant marker of drug resistance, or arguing that in silico modeling is an effective way for screening combinational drugs? If the authors’ conclusion is related to the finding of an AURKA-inhibitor as a new combinational drug for KRAS inhibitors, more in vitro evidence should be presented in the manuscript, and a separate manuscript would be needed.

Whatever is the conclusion or hypothesis, the results should be reorganized or some parts should be removed. After their conclusion is made clear, results and discussion should be focused on the issues raised by the hypothesis. Lengthy discussion which is not focused on the main argument can distract readers.

The significance of Figure 5, which seems to be relevant to the authors’ major argument, is not well described in the result section.

The differential results between 2D and 3D cultures in drug resistance are not well described in the result section. Authors seems to argue that the results from 3D is more reliable. However, the clinical evidence is not clearly presented, and the results from 3D seem to be still questionable. Clinical evidence which can support the hypothesis should be presented at the result section, and the presentation of animal data, if available or reported, will enhance the authors’ argument.

In addition, there are some inconsistency or typos.

  1. Lines 44 and 607: The sentences should be described consistently: EMT is more a marker for than a marker of drug resistance. Or, EMT is more a marker of than a marker for drug resistance.

 In vitro and in silico in the manuscript should be italicized.

Author Response

Coverletter -Revisions

Name: Gudrun Dandekar and Thomas Dandekar

Affiliation: Chair of Tissue Engineering and Regenerative Medicine, University Hospital Würzburg, Röntgenring 11, Würzburg 97070 and Department of Bioinformatics, Biocenter, University of Würzburg, Am Hubland, Würzburg 97074, Germany

Email: gudrun.dandekar@uni-wuerzburg.de, dandekar@biozentrum.uni-wuerzburg.de

Dear Reviewer:

We send you here our revised version of our manuscript “EMT, stemness and drug resistance in biological context: a 3D tumor tissue/ in silico platform for analysis of combinatorial treatment in NSCLC with aggressive KRAS-biomarker signatures” for your special issue on "Tumor Models and Drug Targeting In Vitro".

Thank you for reviewing our manuscript. We incorporated all comments, re-structured the manuscript and focused our hypothesis accordingly. We added further experiments, including differential expression of c-MYC in our KRASG12C-models and the correlation of the efficacy of the described combination therapy with expression of this transcription factor on our 3D tumor models. We also confirmed no relevant toxicity of the AURKA-inhibitor alisertib in fibroblasts. Furthermore, we integrated references regarding AURKA-inhibitor clinical trials and MDR expression in HCC44 cells. The combinatorial treatment of the allel-specific KRAS-G12C-inhibitor with alisertib in highly invasive and resistant HCC44 cells with EMT illustrates as a proof-of-principle how in vitro results go hand in hand with in silico analysis to identify biomarker signatures in lung cancer. Moreover, we revised the manuscript for spelling mistakes and style.

We attach a point-by-point list for the changes performed in the manuscript for your convenience.

Sincerely yours

Thomas Dandekar and Gudrun Dandekar

Point-by-Point Checklist:

Comments by Reviewer 1:  

Although I agree that the significance of EMT for drug resistance is still in debate, the evidences for the authors’ conclusion in the manuscript are not well presented. The results should be rearranged more consistently according to their hypothesis or argument.

We agree that the in silico part was too isolated from the in vitro experiments, leading to confusion about the actual role of the in silico topology. The results part was therefore restructured. The in silico network models and supports by its network logic (Boolean network) the findings of the in vitro experiments. Therefore, we moved the in silico topology and the simulations of the models after treatment with ARS-1620 to the part right behind the drug testing in vitro (now Figure 5). The simulations of combinatorial treatments, supporting the results of the in vitro findings, are now summarized in one single Figure (Figure 8).

The significance of the authors’ conclusion from an in silico model is not well described. Is it supporting that EMT is not a significant marker of drug resistance, or arguing that in silico modeling is an effective way for screening combinational drugs? If the authors’ conclusion is related to the finding of an AURKA-inhibitor as a new combinational drug for KRAS inhibitors, more in vitro evidence should be presented in the manuscript, and a separate manuscript would be needed.

Thank you for your helpful advice to better explain the role of the in silico model. The in silico model integrates EMT as a systemic output similar to apoptosis and proliferation. Signaling which leads to EMT is interconnected to several cascades. Changes of EMT levels can be observed but the model does not support EMT as a marker or maker of drug resistance in itself and serves primarily as an effective screening tool for combinatorial drugs. To integrate EMT in simulations correctly, the KRAS-inhibitor non-responder cell line HCC44 has to show a higher EMT than the responder cell line H358. This is shown in the new Figure 5 B as mentioned above. The in silico model was established according to the in vitro experiments with three combinatorial targeted treatments with the KRAS-inhibitor: SHP2-inhibitor, crizotinib and gefitinib are suggested in the literature to serve as controls to optimize network connections in the signaling topology. After this, the system is now able to correctly predict as a proof-of-concept the outcome of the combinatorial treatment with the AURKA-inhibitor simultaneously in vitro and in silico. To clarify this, we explain now in chapter 3.8. that the three treatments with SHP2-inhibitor, crizotinib, gefitinib were found in the literature and served as controls to set up the in silico model correctly. Furthermore, we explain why it makes sense to test the AURKA inhibitor specifically in HCC44 from the network topology perspective. HCC44 cells display besides a high EMT level, KRAS, P53, LKB1and KEAP1 mutations and also a high c-MYC expression. AURKA shows a strong connection to EMT, LKB1 and c-MYC in the network. We inserted a supplemental figure (Figure S4) which shows the higher c-MYC expression in HCC44 compared to H358 in 3D culture conditions as expected from a database search, which we also reference (Ref 32 and 33). We further added more in vitro evidence for the suggested combination strategy using the PDX derived cell lines with differential c-MYC expression in a new supplemental Figure S7. We inserted the protocol for the Western Blot experiment in the Material and Methods section (lines 286 to 302).

Whatever is the conclusion or hypothesis, the results should be reorganized or some parts should be removed. After their conclusion is made clear, results and discussion should be focused on the issues raised by the hypothesis. Lengthy discussion which is not focused on the main argument can distract readers.

We thank you for your well-taken suggestions. Our hypothesis is that EMT is not a consistent predictor for drug response. Instead, we propose that the biomarker signature of the cell line is an important predictor for invididualized therapies by targeting tumor-specific cascades in a complex network. This is now clearly stated already at the Introduction. Furthermore, we focussed the results and reorganized the paper accordingly: We removed the experiments with galunisertib, revised the abstract, reorganized the results sections, and shortened the discussion to focus on our hypothesis. Detailed in silico methods explanations were shifted to the supplement (Table S2). Our combined in silico / in vitro results are a proof-of-concept that understanding the tumor cell-specific network and its different pathway dependencies on the biomarker signature is the key for a rational combinatorial targeted treatment while individual markers such as EMT do not correlate so well with the cellular systems response and treatment success.

The significance of Figure 5, which seems to be relevant to the authors’ major argument, is not well described in the result section.

We agree that the results in Figure 5 were not described in much detail while it is of course an important experiment supporting our findings. Accordingly, we explained the TGF-β part now in an own chapter 3.5. and described the changes in proliferation and apoptosis in more detail.

The differential results between 2D and 3D cultures in drug resistance are not well described in the result section. Authors seems to argue that the results from 3D is more reliable. However, the clinical evidence is not clearly presented, and the results from 3D seem to be still questionable. Clinical evidence which can support the hypothesis should be presented at the result section, and the presentation of animal data, if available or reported, will enhance the authors’ argument.

Thank you for these comments. In paragraph 3.7 we included now a section where we explain in more detail, why we think our models deliver a more reliable drug response prediction and inserted also references. Furthermore, we integrated references showing ongoing clinical studies with AURKA inhibitors, specifically also with alisertib (discussion, line 829 to 831). We inserted a reference showing the relevance of AURKA expression for prognosis in lung cancer patients and in resistance development in KRAS-mutated lung cancer cells (discussion, line 833).

In addition, there are some inconsistency or typos.

Lines 44 and 607: The sentences should be described consistently: EMT is more a marker for than a marker of drug resistance. Or, EMT is more a marker of than a marker for drug resistance.

In the short summary (line 44), in the title of Figure 6 (line 578) and in the beginning of the discussion (line 692) sentences were changed to a consistent form: “EMT is more a marker of drug resistance than a maker”

In vitro and in silico in the manuscript should be italicized.

We italicized such terms throughout the manuscript. Moreover, we went over the whole manuscript once again checking for typos, language and style.

Reviewer 2 Report

Multicellular 3D in vitro systems can overcome the limits of 2D culture and bridge the gap between experimental tractability and physiological relevance. Mechanical and biochemical cues are critical for cancer formation, including shape, cell-cell/cell–ECM interactions, tissue stiffness, and particular gradients, which can be reproduced using 3D models. Recent advances in 3D cancer models offer the potential to improve drug discovery and testing platforms and enable the development of personalized cancer treatments. In this article, the authors demonstrate the 3D tumor tissue/ in silico platform for analyzing combinatorial treatment in NSCLC with aggressive KRAS-biomarker signatures. The authors have produced several positive outcomes from this study. However, I have a few concerns from this article.

What criteria were used to include and exclude the patient tumor sample? That must be stated in the text by the author.

How long was the culture kept for the development of resistant cells (HCCresA1, HCCresA2, HCCresA3), and what were the IC50 values?

To ensure the cytocompatibility of Aurora Kinase A inhibitor in combinatorial drug treatment, a normal cell line should also be included in the study.

I recommend that the authors run the multidrug resistance (MDR) protein to confirm the drug resistance.

In lung cancer, how can homeostasis be altered by combinatorial treatment? Discuss

PCR array for lung cancer. The number of repeats in the array plates is unknown. I recommend including a table of PCR array plate data in the supplementary file.

Author Response

Coverletter -Revisions

Name: Gudrun Dandekar and Thomas Dandekar

Affiliation: Chair of Tissue Engineering and Regenerative Medicine, University Hospital Würzburg, Röntgenring 11, Würzburg 97070 and Department of Bioinformatics, Biocenter, University of Würzburg, Am Hubland, Würzburg 97074, Germany

Email: gudrun.dandekar@uni-wuerzburg.de, dandekar@biozentrum.uni-wuerzburg.de

Dear Reviewer:

We send you here our revised version of our manuscript “EMT, stemness and drug resistance in biological context: a 3D tumor tissue/ in silico platform for analysis of combinatorial treatment in NSCLC with aggressive KRAS-biomarker signatures” for your special issue on "Tumor Models and Drug Targeting In Vitro".

Thank you for reviewing our manuscript. We incorporated all comments, re-structured the manuscript and focused our hypothesis accordingly. We added further experiments, including differential expression of c-MYC in our KRASG12C-models and the correlation of the efficacy of the described combination therapy with expression of this transcription factor on our 3D tumor models. We also confirmed no relevant toxicity of the AURKA-inhibitor alisertib in fibroblasts. Furthermore, we integrated references regarding AURKA-inhibitor clinical trials and MDR expression in HCC44 cells. The combinatorial treatment of the allel-specific KRAS-G12C-inhibitor with alisertib in highly invasive and resistant HCC44 cells with EMT illustrates as a proof-of-principle how in vitro results go hand in hand with in silico analysis to identify biomarker signatures in lung cancer. Moreover, we revised the manuscript for spelling mistakes and style.

We attach a point-by-point list for the changes performed in the manuscript for your convenience.

Sincerely yours

Thomas Dandekar and Gudrun Dandekar

Point-by-Point Checklist:

Comments by Reviewer 2:

What criteria were used to include and exclude the patient tumor sample? That must be stated in the text by the author.

In Material and Methods in the section “Patient tumor samples” (line 200/201) the following sentence was included for clarification: “All available adenocarcinoma samples were included into the study.”

How long was the culture kept for the development of resistant cells (HCCresA1, HCCresA2, HCCresA3), and what were the IC50 values?

Resistance towards gefitinib developed in HCCresA2 and HCCresA3 after about 4 months of permanent culture with 1 µM gefitinib. Thus, we integrated in the Materials and Methods section (line 175 to 178) the sentence: “HCCresA2 and HCCresA3 cells got resistant after about 4 months of permanent treatment with 1 µM gefitinib, whereas HCCresA1 did not develop a complete resistance in this time-period.” In the result section, we inserted “…after about four months treatment” in line 448 to 449.

We agree with the inconsistency that IC50 values were indicated for the PDX-derived cell lines, but neither for HCCresA1, HCCresA2, HCCresA3 nor for H358 and HCC44 cells in the supplements. Therefore, we added the corresponding IC50-values to the supplemental Figure S2. However, the IC50s were not reached with the applied concentrations in the case of HCCresA1, HCCresA2, HCCresA3 and we hence defined them as IC50>10 µM.

To ensure the cytocompatibility of Aurora Kinase A inhibitor in combinatorial drug treatment, a normal cell line should also be included in the study.

Thank you for your suggestion! We added in the supplemental Figure (S5) the testing of fibroblasts as normal cells and confirm here no relevant toxicity of alisertib in primary lung fibroblasts. Furthermore, we included a reference on AURKA inhibitors in clinical trials in the discussion (line 831), where it is shown that alisertib reached as only AURKA-inhibitor phase III.

I recommend that the authors run the multidrug resistance (MDR) protein to confirm the drug resistance.

We apologize for not showing any experiments on this topic. We integrated instead in the discussion section (line 796) a link on a suitable data base and confirmed by a search in the data base that there is no significant expression of MDR and ABC transporters in HCC44 cells reported (http://celllines.tron-mainz.de/).

In lung cancer, how can homeostasis be altered by combinatorial treatment? Discuss

Thank you for this question. We inserted now in the results section in paragraph 3.7 the following : “Homeostasis is the equlibrium of the cell. In a healthy cell, the differentiation pathways reliably help the cell to carry out its specific functions. In lung cancer the system state which is actively preserved in the cancer cell is high proliferation, low differentiation and low apoptosis. A combinatorial treatment allows to rectify this system state by combining drugs targeting specific pathways, for instance a kinase inhibitor blocking proliferation with another drug stimulating apoptosis.” (lines 617 to 622)

PCR array for lung cancer. The number of repeats in the array plates is unknown. I recommend including a table of PCR array plate data in the supplementary file.

A copy of all investigated genes is now integrated (line 313) as a separate supplementary file: Gene_Table.xls. Here all wells of a single 96-well PCR array plate are described in detail. In the PDF file the table is integrated in line 1266.

Round 2

Reviewer 1 Report

The authors’ main arguments are that EMT is not a significant drug resistant marker in KRAS mutant lung cancer cells, and that the addition of AURKA inhibitors to KRAS inhibitor treatment could be the most promising drug combination for KRAS mutant lung cancer patients.

The manuscript is acceptable if the following typos are corrected.

  1. ‘In vitro’ and ‘in silico’ should be italicized.
  2. Line 106, 1.500 should be 1,500.
  3. Line 549, ‘andInvasion’ needs a space.

Author Response

Name: Gudrun Dandekar and Thomas Dandekar

Affiliation: Chair of Tissue Engineering and Regenerative Medicine, University Hospital Würzburg, Röntgenring 11, Würzburg 97070 and Department of Bioinformatics, Biocenter, University of Würzburg, Am Hubland, Würzburg 97074, Germany

Email: gudrun.dandekar@uni-wuerzburg.de, dandekar@biozentrum.uni-wuerzburg.de

Dear Reviewer:

We send you here our further revised version of our manuscript “EMT, stemness and drug resistance in biological context: a 3D tumor tissue/ in silico platform for analysis of combinatorial treatment in NSCLC with aggressive KRAS-biomarker signatures” for your special issue on "Tumor Models and Drug Targeting In Vitro".

Thank you for re-reviewing our manuscript. We incorporated all reviewer comments, further specified the included patient group, integrated detailed information on tested concentrations and resistance and specified and better explained the term “resistant HCC44 cells”.

We attach a point-by-point list for the changes performed in the manuscript for your convenience and hope they will proof satisfactory.

Sincerely yours

Thomas Dandekar and Gudrun Dandekar

Point-by-Point Checklist:

Comments by Reviewer 1:

  1. ‘In vitro’ and ‘in silico’ should be italicized.

--A (Answer): Yes, we agree and apologize for this, the formatting was inadvertently reversed by the submission process. This has been corrected in the revision of this manuscript. However, we cannot be sure that again the system will enforce a formatting without italics. In that case, we will again correct this in the proofs.

  1. Line 106, 1.500 should be 1,500.

--A: Thank you, we corrected this as requested according to the MDPI guidelines.

  1. Line 549, ‘andInvasion’ needs a space.

--A: Thank you, this has been corrected as well as other typos.

Reviewer 2 Report

In the updated manuscript, the authors have made significant improvements. However, there are some concerns that have not been adequately addressed.

1. The authors failed to show the precise criteria for patient sample exclusion and inclusion. According to the authors "All available adenocarcinoma samples were included in the study,", is insufficient information about the sample. Race, ethnicity, disease type and stage, as well as age, gender, previous treatment history, and the presence or absence of other medical disorders, should all be taken into account

2. The statement regarding the IC50 value is very confusing. The authors should include the clear data of IC50 values in the text, there is no means of defining >10 µM.

3. Although MDR expression in HCC44 was not found in the database (http://celllines.tron-mainz.de/), the authors should explore additional drug resistance indicators in the cells. The authors should show the evidence of resistance as they were working with resistance cells.

Author Response

Coverletter for our Revisions

Name: Gudrun Dandekar and Thomas Dandekar

Affiliation: Chair of Tissue Engineering and Regenerative Medicine, University Hospital Würzburg, Röntgenring 11, Würzburg 97070 and Department of Bioinformatics, Biocenter, University of Würzburg, Am Hubland, Würzburg 97074, Germany

Email: gudrun.dandekar@uni-wuerzburg.de, dandekar@biozentrum.uni-wuerzburg.de

Dear Reviewer:

We send you here our further revised version of our manuscript “EMT, stemness and drug resistance in biological context: a 3D tumor tissue/ in silico platform for analysis of combinatorial treatment in NSCLC with aggressive KRAS-biomarker signatures” for your special issue on "Tumor Models and Drug Targeting In Vitro".

Thank you for re-reviewing our manuscript. We incorporated all reviewer comments, further specified the included patient group, integrated detailed information on tested concentrations and resistance and specified and better explained the term “resistant HCC44 cells”.

We attach a point-by-point list for the changes performed in the manuscript for your convenience and hope they will proof satisfactory.

Sincerely yours

Thomas Dandekar and Gudrun Dandekar

Point-by-Point Checklist:

Reviewer 2:

  1. The authors failed to show the precise criteria for patient sample exclusion and inclusion. According to the authors "All available adenocarcinoma samples were included in the study,", is insufficient information about the sample. Race, ethnicity, disease type and stage, as well as age, gender, previous treatment history, and the presence or absence of other medical disorders, should all be taken into account.

--A: The stainings used in the manuscript from paraffin embedded tumor material were used as references for our models how they correlate to findings in the patient. All patient material had been pseudonymized already in the hospital, and treatment history was in most cases not transferred to the lab as we investigated in this study only the correlation between EMT and invasion. Thus, pre-treatment was no exclusion criterion. Nevertheless, changes of EMT-markers upon treatment will be an important point in further studies. Race, ethnicity, and other disorders were not reported. To clarify these points, we inserted now in the Materials and Methods part in the section “Patient tumor samples” the sentence: “Patient material was derived from 10 NSCLC adenocarcinoma patients older than 18 years from both sexes. Ethnicity, previous treatments, and other medical disorders were neglected.” (Lines 199 to 201)

  1. The statement regarding the IC50 value is very confusing. The authors should include the clear data of IC50 values in the text, there is no means of defining >10 µM.

We agree, and removed">10 µM” from all figures, supplemented the paragraf “Cells” in the Materials and Methods section with further explanations, and referenced our definition of resistance: “While HCCresA2 and HCCresA3 cells displayed resistance towards the treatment with gefitinib in concentrations up to 10 µM, HCCresA1 cells still exhibited an intermediate sensitivity in comparison (Figure S2). HCCresA2 and HCCresA3 cells displayed reduced sensitivity towards EGFR inhibition after about 4 months of permanent treatment with 1 µM gefitinib, whereas HCCresA1 still showed a partial sensitivity in this time-period. Regarding gefitinib sensitivity, we are referring to Noro et al. cells with an IC50 lower than 1 µM as highly sensitive, cells with an IC50 between 1 µM and 10 µM as intermediate-sensitive and cells with an IC50 greater than or equal to 10 µM as resistant, respectively [21].” (Lines 167 to 176)

(Reference 21: Noro, R., Gemma, A., Kosaihira, S., Kokubo, Y., Chen, M., Seike, M., Kataoka, K., Matsuda, K., Okano, T., Minegishi, Y., Yoshimura, A., & Kudoh, S. (2006). Gefitinib (IRESSA) sensitive lung cancer cell lines show phosphorylation of Akt without ligand stimulation. BMC cancer, 6, 277)

  1. Although MDR expression in HCC44 was not found in the database (http://celllines.tron-mainz.de/), the authors should explore additional drug resistance indicators in the cells. The authors should show the evidence of resistance as they were working with resistance cells.

We apologize to be here not sufficiently clear. We now removed throughout the text “highly resistant” HCC44 cells and changed it where applicable to more careful wording such as “more resistant” or “higher resistance in KRASG12C-mutated HCC44 cells compared to H358” as this is otherwise obviously misleading. (See: abstract, introduction, results: section 3.7, discussion section 4.3: here, we are now also more carfully in the wording of this sentence (Line 770/771): “To find hints that higher resistance in HCC44 cells is not only based on multi drug resistance (MDR) proteins, we checked the expression of MDR and ABC transporters in an online available data base [74,75] and did not find any expression

Round 3

Reviewer 2 Report

All of the comments have been meticulously addressed by the authors. It is feasible to accept the manuscript in its current state.